# RepQA: Evaluating Readability of Large Language Models in Patient Education Question Answering

**Weikang Qiu**[1,*,†]  **Tinglin Huang**[1,*]  **Ryan Rullo**[2]  **Giselle OConnor**[2]  **Yuchen Kuang**[4]

**Ali Maatouk**[1]  **S. Raquel Ramos**[2,3]  **Rex Ying**[1]

[1]Yale University, Department of Computer Science**,**
[2]Yale University, School of Nursing,
[3]Yale University, School of Public Health, [4]Northeastern University

## Abstract

Large language models (LLMs) have shown great promise in addressing complex medical and clinical problems. However, while most prior studies focus on improving accuracy and reasoning abilities, a significant bottleneck in developing effective healthcare agents lies in the readability of LLM-generated responses, specifically, their ability to answer patient education problems clearly to people without a medical background. In this work, we introduce RepQA[1], a benchmark designed to systematically evaluate the readability of LLMs in patient education question-answering (QA). RepQA comprises 533 expert-reviewed QA pairs sourced from 27 online resources, reflecting common concerns of lay users across 4 categories. RepQA incorporates a proxy multiple-choice QA task to directly assess the informativeness of generated responses, alongside two readability metrics. We present a comprehensive study of 25 LLMs on RepQA, evaluating both instruction-following (answering at requested reading levels) and readability understanding (recognizing the level of questions and texts). We find that current models fail to achieve target levels and struggle to identify appropriate reading levels, revealing a gap between raw reasoning ability and effective communication. We then compare four approaches for improving readability: standard prompting, chain-of-thought prompting, Group Relative Policy Optimization (GRPO), and a token-adapted GRPO. While readability-aware post-training substantially improves readability metrics, it often reduces QA accuracy due to over-simplification, exposing a significant readability–accuracy trade-off. These findings point toward methods for building more user-centered public health agents.

## 1 Introduction

Benefiting from extensive training on large-scale corpora, large language models (LLMs) have demonstrated remarkable reasoning capabilities in solving complex problems OpenAI et al. (2024a); DeepSeek-AI et al. (2024); Team et al. (2024b); Qiu et al. (2025); Zhao et al. (2023). One of the most impactful application areas is healthcare, which demands strong domain-specific knowledge and a high reliability standard Singhal et al. (2023); Thirunavukarasu et al. (2023); Nazi & Peng (2024). Previous studies have introduced benchmarks Yao et al. (2024); Jin et al. (2019); Jiang et al. (2024); Wang et al. (2024b); Jin et al. (2019) targeting different levels of medical reasoning tasks.

Despite their promising performance, the **readability** of LLM-generated responses, an essential factor for patient education, i.e., communicating health information to laypeople, has been largely

---

*Equal contribution.
†Correspondence to `weikang.qiu@yale.edu`
[1]Code and dataset available at `https://anonymous.4open.science/r/RepQA-45A8/`

Figure 1: Two examples of user interactions with low- and high-readability patient education agents. Medical jargon is underscored in the responses generated by the low-readability agent.

Table 1: The comparison between REPQA and the existing healthcare-related benchmark datasets.

| Dataset | Answer Format | Sources of Questions | Evaluation Metrics | Target Audience |
|---------|---------------|----------------------|--------------------|-----------------|
| MedQA Jin et al. (2021) | Multiple Choice | Medical Examination | Accuracy | Medical Professionals |
| MedMCQA Pal et al. (2022) | Multiple Choice | Medical Examination | Accuracy | Medical Professionals |
| MMLU Hendrycks et al. (2020) | Multiple Choice | Medical Examination | Accuracy | Academic |
| PubMedQA Jin et al. (2019) | Multiple Choice | PubMed Abstract | Accuracy | Clinical Researchers |
| LiveQA Abacha et al. (2017) | Long Answer | National Library of Medicine | Human Evaluation | Consumers |
| MedicationQA Abacha et al. (2019) | Long Answer | MedlinePlus & DailyMed | Human Evaluation | Medicine Consumers |
| JAMA & Medbullets Chen et al. (2025) | Long Answer & Multiple Choice | JAMA Challenge & USMLE | NLG Metrics & Accuracy | Medical Professionals |
| REPQA | Long Answer & Multiple Choice | Public Health FAQs & Experts | Readability & Accuracy | Lay Users / Patients |

overlooked and remains systematically underexplored. As shown in Figure 1, users without a medical professional background may find low-readability responses filled with medical jargon difficult to comprehend, whereas a plain-language answer can convey the same essential information more accessibly. Moreover, state-of-the-art LLMs are increasingly optimized for complex reasoning Shao et al. (2024), raising the question of whether these gains come at the expense of readability. More advanced reasoning may lead to more technical responses that are harder for lay users to understand.

In this study, we introduce **REPQA** to conduct a comprehensive investigation into the **RE**adability of LLMs in **P**atient **E**ducation question-**A**nswering (QA) tasks. A key strength of REPQA lies in its expert-grounded foundation: it comprises of high-quality test set with 533 QA pairs curated from 27 online public health resources, all reviewed by professionals in nursing, public health, and clinical communication. Different from prior related benchmarks (Table 1), REPQA focuses on frequently asked public health questions from lay users or patients, and introduces a novel proxy multiple-choice QA that directly evaluates the accuracy of generated responses, rather than the underlying factual knowledge of the LLMs. Besides, we collect a large-scale corpus of 36,060 public health questions to serve as a post-training dataset, supporting the exploration of readability-enhancement strategies.

We evaluate 25 LLMs on REPQA using two readability metrics (Flesch-Kincaid grade level Kincaid et al. (1975) and professional score) and one accuracy metric based on our proxy multiple-choice QA. Our study targets three capabilities: (i) answering at a specified reading level, (ii) inferring the appropriate level for a question, and (iii) using readability as a reward signal. We find that current models mostly miss target levels and poorly identify appropriate levels, limiting their applicability for public-health agents. We then benchmark four readability-improvement strategies: two test-time methods (standard prompting Ribeiro et al. (2023); Malik et al. (2024), chain-of-thought Wei et al. (2022)) and two post-training methods (GRPO Shao et al. (2024) and a token-adapted GRPO). Readability-aware post-training yields large gains in readability metrics but reduces QA accuracy due to over-simplification, underscoring a difficult readability–accuracy trade-off. These findings offer insights for future work on developing trustworthy healthcare agents and may be generalized to other educational domains Wang et al. (2024a).

In summary, our contributions are listed as follows:

- We introduce REPQA, a novel benchmark for patient education QA aimed at lay users, with two readability metrics (Flesch–Kincaid grade and professional-term proportion) and a proxy multiple-choice QA protocol for accuracy.

- We conduct a comprehensive study of 25 LLMs on REPQA, revealing that most models struggle to (i) generate at requested reading levels and (ii) identify an appropriate level for a given question.

- We explore four existing strategies to improve LLM readability and propose a novel method, the token-adapted GRPO method. Readability improves substantially, but often at an accuracy cost, highlighting a challenging readability–accuracy trade-off.

## 2    RELATED WORK

**Evaluation of LLM on healthcare QA**   To evaluate the capabilities of LLMs in solving healthcare and medical problems, multiple benchmarks have been curated from diverse sources and presented in various formats Singhal et al. (2023); Liévin et al. (2024); Thirunavukarasu et al. (2023). For instance, MedQA Jin et al. (2021), MedMCQA Pal et al. (2022), MMLU Hendrycks et al. (2020); Wang et al. (2024b), and PubMedQA Jin et al. (2019) contain multiple-choice questions sourced from medical licensing examinations or PubMed articles, while LiveQA Abacha et al. (2017) and MedicationQA Abacha et al. (2019) focus on long-form question answering for medicine-related problems. Recent efforts have shifted toward more complex clinical tasks requiring multi-step reasoning and decision-making Yao et al. (2024); Chen et al. (2025); Nori et al. (2024); Saab et al. (2024), as well as incorporating additional modalities such as electronic health records Ou et al. (2025); Bardhan et al. (2022); Kweon et al. (2024), radiology Zuo et al. (2025); Liu et al. (2021), and pathology He et al. (2020); Lu et al. (2024). Besides, several studies explore the reliability of LLMs with retrieval augmented generation Xiong et al. (2024). Our proposed benchmark, REPQA, is the first to comprehensively evaluate the readability of LLM-generated responses to healthcare-related questions, using a dataset composed of publicly available medical queries from lay users, rather than professional exam questions or specialized clinical tasks.

**Readability control in LLMs**   There are three main strategies for controlling the complexity or reading level of text generated by large language models. (1) Prompt- or instruction-based methods modulate output difficulty by varying instructions or providing exemplars Tran et al. (2024); Malik et al. (2024); Barayan et al. (2024); Pu & Demberg (2023); (2) Fine-tuning-based methods, such as Chi et al. (2023) fine-tuned language model on same-level paraphrasing datasets, and Ribeiro et al. (2023) employed PPO Schulman et al. (2017) with the measured reading level Kincaid et al. (1975) as a reward signal to optimize the models; (3) Decoding-based methods incorporate constraints during text generation Ribeiro et al. (2023); Luo et al. (2022). Several studies investigated the readability of LLMs' generated responses regarding certain cancers Hershenhouse et al. (2024); Gencer (2024) or the patient education handouts Swisher et al. (2024). While most prior studies focus on text summarization, REPQA specifically targets the readability of LLM in patient education QA tasks.

## 3    REPQA BENCHMARK

In this section, we introduce REPQA, which comprises a high-quality collection of patient education QA pairs with corresponding multiple-choice questions for evaluation, as well as a large-scale set of questions for training. We detail the dataset curation pipeline in Section 3.1, outline the evaluation metrics in Section 3.2, and describe the studied problems in Section 3.3.

### 3.1    DATASET CURATION

**Expert-Grounded Patient Education QA**   We collected a diverse set of publicly available healthcare-related QA pairs from 27 authoritative online resources covering a broad range of public health topics, including general health, cardiovascular disease, cholesterol, diabetes, HIV, diet, exercise, alcohol, smoking, sleep, mental health, sexual health, and health equity. Sources were selected for their accessibility to lay audiences and their coverage of frequently asked questions by the general public. All the links can be found in Appendix A.1.

For the evaluation dataset, domain experts further manually reviewed and filtered out QA pairs containing overly technical content, following national, authoritative, evidence-based research and clinical guidelines Doody & Noonan (2016); Chiang-Hanisko et al. (2016); Corry et al. (2013). Questions are phrased in plain language and do not include user profiles. A detailed description of

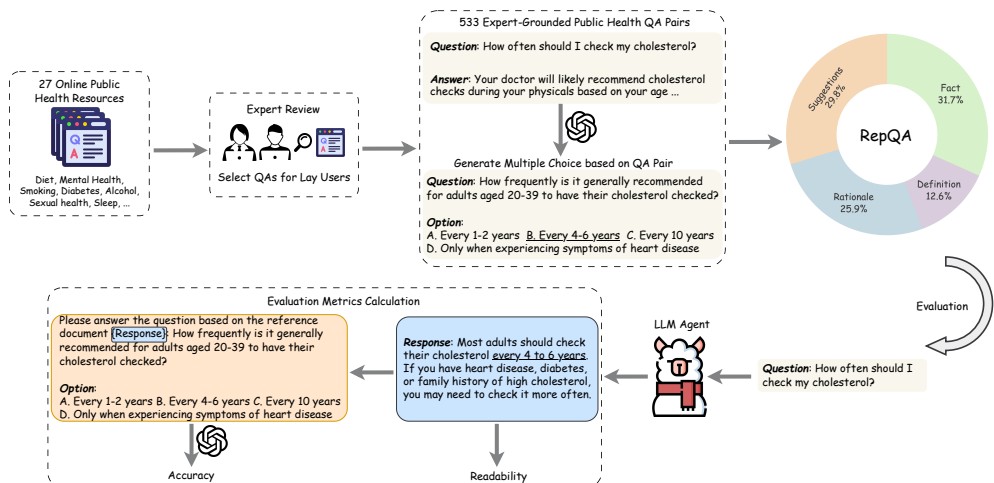

Figure 2: Overview of the REPQA dataset construction and evaluation pipeline. The dataset is curated through expert review and multiple-choice question generation, while the evaluation involves readability assessment and accuracy measurement.

Table 2: Examples and distribution of question categories in REPQA. Each question is accompanied by a corresponding reference answer.

| Category | Example & Reference Answer | Proportion |
|---|---|---|
| Suggestion | **Q:** How can I quit smoking? 
 **A:** Commit to a quit day, choose a method for quitting (cold turkey or weaning down), talk to your doctor about medications to help quit, make a plan for when you quit on how to resist temptations, remove triggers, and have replacement activities, and then quit on your quit day! | 29.8% |
| Fact | **Q:** What are the chemicals in cigarettes? 
 **A:** There are over 7,000 chemicals in cigarettes including arsenic, formaldehyde, tar, nicotine, and carbon monoxide, all of which are dangerous to your health. | 31.7% |
| Definition | **Q:** What is the Mediterranean Diet? 
 **A:** A Mediterranean diet is considered a healthy option as it emphasizes fruits and vegetables, whole grains, beans and legumes, low-fat dairy products, fish and poultry, and vegetable oils. It also limits sugars, processed foods, and fatty meats. | 12.6% |
| Rationale | **Q:** Why is blood pressure called a "silent killer"? 
 **A:** High blood pressure is sometimes called a silent killer because it can have no physical symptoms; however, the high blood pressure can damage arteries risking serious health events like heart attack or stroke. | 25.9% |

the curation pipeline and the reviewing team's composition is provided in Appendix A.2. As shown in Table 2, the evaluation dataset includes 533 QA pairs with the following categories:

- *Suggestion*: Questions seeking advice or recommended actions. These responses must be highly readable to ensure that users can easily follow health recommendations without ambiguity.

- *Fact*: Questions requesting objective, verifiable information. Presenting facts in lay-accessible language is essential for avoiding misinterpretation.

- *Definition*: Questions asking for explanations of medical terms or conditions. Definitions require especially high readability, as they often introduce unfamiliar terminology.

- *Rationale*: Questions seeking reasons or justifications behind medical guidance. These responses must balance clarity with accuracy in plain language to build user understanding and trust.

**Proxy Multiple Choice QA**    Each question in the final evaluation dataset is paired with a reference answer. To assess the accuracy of generated responses, prior studies have either employed LLMs as judges to rate the relevance of a response against the reference answer Gu et al. (2024), or directly adopted multiple-choice questions Singhal et al. (2023); Liévin et al. (2024). However, the former approach raises concerns about objectivity, especially in the healthcare domain, while the latter assesses the model's direct knowledge rather than the accuracy of its generated explanations.

Table 3: Examples and distribution of question categories in the collected training datasets. A single question can be categorized into multiple categories (e.g., Diet and Weight: How do dietary habits influence obesity according to research findings?)

| Category | Example | Proportion |
|---|---|---|
| Sleep | How does sleep apnea during pregnancy potentially affect newborns? | 8.1% |
| Exercise | What are some effective strategies for improving family health through physical activities? | 6.3% |
| Smoking | What are some effective strategies individuals can use to quit smoking and support a smoke-free lifestyle? | 4.5% |
| Weight | How is body mass index (BMI) calculated, and what factors can affect its accuracy? | 7.2% |
| Diet | How can someone maintain a healthy diet while dining out? | 10.3% |
| HIV Care | How can gay men effectively manage their risk of HIV? | 14.4% |
| Mental Health | How can parents support the mental health of their children? | 12.6% |
| Cardiovascular | What is the primary difference between cardiac arrest and a heart attack? | 18.7% |
| Diabetes | How do BMI categories relate to the risk of type 2 diabetes across different age groups? | 5.2% |
| Cholesterol | What is the significance of LDL and HDL cholesterol in relation to heart health? | 5.1% |
| SOGI | How can schools and communities support LGBTQI+ youth to create a safer environment? | 12.4% |
| Others | What are the potential benefits of utilizing specialized healthcare models in patient treatment? | 33.7% |

To bridge this gap, we propose a proxy multiple-choice selection task, where a generated answer serves as a reference document for an `LLM-Critic` tasked with selecting the correct option, as illustrated in Figure 2. The multiple-choice QA is generated by an LLM (`LLM-Proposer`) using each question and its reference answer. Higher accuracy suggests that the response captures the essential information, making it a reliable proxy for factual correctness.

In practice, we observed that the initial questions generated by the `LLM-Proposer` were often too easy, and that the `LLM-Critic` simply relied on its own knowledge rather than the provided context. To overcome these issues, we integrate the `LLM-Critic` into the problem generation process, using it as a feedback mechanism to iteratively refine the multiple-choice questions, ensuring that they are non-trivial and that their solutions rely on the provided context. In practice, we use GPT-4.1 as the `LLM-Proposer` and GPT-4o-mini as the `LLM-Critic`. The complete algorithm is detailed in Algo. 1, and the prompts for both LLMs are included in Appendix B.2.

**Large Scale Healthcare Questions**  In addition to the evaluation QA pairs, we curated a large-scale collection consisting of 36,060 public health questions. These queries span an extensive range of public health topics, including sleep, exercise, smoking, weight, diet, HIV care, mental health, cardiovascular, diabetes, cholesterol, and SOGI, from 55 publicly available resources (Appendix A.1). The data sources were manually selected by nursing and public health professionals, following the same guidelines used in curating the evaluation dataset. Summary statistics and representative examples for each category are presented in Table 3. Notably, this dataset includes only questions without reference answers, and allows the exploration of different training strategies, such as GRPO Shao et al. (2024), with readability metrics serving as verifiable rewards.

## 3.2  EVALUATION METRICS

**Flesch-Kincaid Grade Level**  We use the Flesch-Kincaid grade level Kincaid et al. (1975) as one of our readability metrics, which is an established standard for assessing reading difficulty, ranging from kindergarten to post-graduate levels. This metric enables fine-grained analysis of language complexity across model-generated responses. Given a text, the calculation is based on the average number of syllables in a word and the average number of words in a sentence:

$$\pi = 0.39 \left( \frac{\text{Total Words}}{\text{Total Sentences}} \right) + 11.8 \left( \frac{\text{Total Syllables}}{\text{Total Words}} \right) - 15.59 \qquad (1)$$

The intuition is that sentences with many words or words with numerous syllables are generally more difficult to read and understand, corresponding to a higher Flesch-Kincaid grade level. The range of the grade levels can be categorized into different school levels, such as 6-9 corresponds to the middle school level. More details can be found in Appendix C.1.

**Professional Score**  During communication between the agent and lay users without a medical background, the frequency of professional or technical terms can significantly affect the users' reading experience. In light of this, we incorporate the proportion of health-related professional terms in the

text as one of our readability metrics, reflecting the degree of technicality in a response:

$$\rho = 100\% \cdot \frac{\text{Total Professional Terms}}{\text{Total Words}} \tag{2}$$

We compiled a list of professional terms from the Harvard Medical Dictionary[2], resulting in a vocabulary of 2,058 terms. The proposed professional level metric complements the Flesch-Kincaid Grade Level by incorporating domain-specific terminology, which may affect comprehension for lay audiences but is not captured by traditional readability formulas.

**Accuracy via Proxy QA** As described in Section 3.1, we propose using a multiple-choice selection task as a proxy for evaluating the factual accuracy of generated answers. Specifically, the generated answer is used as contextual input to guide an LLM-based critic in selecting the correct option from a set of choices. Accuracy is the proportion of cases where the selected option matches the gold label, averaged over the evaluation set. Higher accuracy indicates that the generated answers convey information more accurately and effectively. An ideal healthcare agent should produce responses that are both easy to understand and highly accurate.

### 3.3 INVESTIGATION ON LLMs' READABILITY

Using our curated REPQA, we study LLMs' ability to adapt, assess, and optimize readability. The prompts can be found in Appendix B.2. We pose three research questions:

**RQ1: Can LLMs generate responses at a specified reading level?** People with different educational backgrounds have varying health literacy. Therefore, healthcare agents should adapt the readability of their answers to a requested level (e.g., 5th grade or middle school). We examine whether LLMs follow explicit readability instructions without sacrificing accuracy. We compare each response's Flesch–Kincaid grade to the requested level and assess accuracy via proxy QA tasks. In addition, we evaluate LLM's readability understanding capability: LLMs are asked to predict the reading level of given responses, and we quantify their accuracy against the Flesch–Kincaid grade.

**RQ2: Can LLMs determine what readability is needed for a given question?** Oversimplifying medical content can omit qualifiers and lose information, while overly technical language reduces accessibility. Thus, selecting an appropriate reading level for each question is critical. We investigate whether an LLM can infer the target level from the question alone (without user metadata), balancing clarity and accuracy. Concretely, we extend each question in REPQA into three variants that differ in required medical knowledge, and evaluate the accuracy of the models' predictions.

**RQ3: Can readability serve as a reward signal?** Reinforcement learning has shown promise for steering LLMs toward preferred behaviors. We study whether readability can act as a useful reward to drive responses toward a requested level without sacrificing content quality. Specifically, the reward function is calculated based on the weighted combination of Flesch-Kincaid grade level and the frequency of professional terms:

$$r = -1 \cdot (\alpha \cdot |\pi - 6| + \beta \cdot \rho + \gamma \cdot \nu) \tag{3}$$

where $\alpha$, $\beta$ and $\gamma$ are hyperparameters used to balance components. $\nu = 1$ if the response is considered trivial and 0 otherwise. A Flesch-Kincaid grade level ($\pi$) closer to 6, i.e., the middle school reading level, and a lower professional term proportion ($\rho$) result in a higher reward. We adopt Group Relative Policy Optimization (GRPO) and a token-adapted variant TA-GRPO that redistributes the sequence-level reward across tokens using weights derived from the occurrence of professional terms. More details regarding TA-GRPO can be found in Appendix C.2.

## 4 EXPERIMENTS

### 4.1 EXPERIMENTAL SETUP

We present a comprehensive evaluation of 25 LLMs, including 12 proprietary models and 13 open-source models. For proprietary models, we consider GPT-4 series OpenAI et al. (2024c), GPT-4o

---

[2]https://www.health.harvard.edu/a-through-c

Table 4: Benchmarking both proprietary and open-source LLMs. Metrics: Grade - Flesch-Kincaid grade level; Prof. - Professional score; Acc. - Accuracy via Proxy QA. More readability metrics are included in Appendix E.

| Model | Suggestion | | | Fact | | | Definition | | | Rationale | | |
|---|---|---|---|---|---|---|---|---|---|---|---|---|
| | Grade | Prof. | Acc. | Grade | Prof. | Acc. | Grade | Prof. | Acc. | Grade | Prof. | Acc. |
| **Proprietary Models** | | | | | | | | | | | | |
| o4-mini | 3.10 | 2.23% | 80.50% | 4.70 | 3.51% | 82.84% | 3.79 | 3.74% | 86.57% | 3.46 | 3.92% | 86.96% |
| o3-mini | 3.58 | 2.22% | 74.21% | 4.17 | 3.26% | 75.74% | 4.56 | 3.95% | 80.60% | 3.99 | 4.16% | 84.06% |
| GPT-4o | 4.30 | 2.25% | 74.21% | 5.27 | 3.22% | 79.29% | **5.61** | 3.73% | **89.55%** | 5.16 | 3.58% | 83.33% |
| GPT-4o-mini | 3.86 | 2.26% | 73.58% | 4.63 | 2.99% | 73.96% | 5.07 | **3.27%** | 85.07% | 4.50 | 3.50% | 79.71% |
| GPT-4.1 | 3.65 | 2.22% | 78.62% | 4.66 | 3.24% | 73.37% | 4.70 | 3.77% | 86.57% | 4.44 | 4.16% | **88.41%** |
| GPT-4 | 4.07 | 2.45% | 75.47% | 4.79 | 3.19% | 70.41% | 4.87 | 3.48% | 85.07% | 4.30 | 3.61% | 78.26% |
| GPT-4-turbo | 4.61 | 2.19% | 72.33% | **5.51** | 2.91% | 65.09% | 5.39 | 3.48% | 76.12% | 5.33 | 3.60% | 78.99% |
| Claude-3.7-Sonnet | 8.95 | 2.49% | **81.32%** | 9.43 | 3.41% | **83.43%** | 7.68 | 3.65% | 85.07% | 7.72 | 3.95% | 84.78% |
| Claude-3.5-Sonnet | **6.68** | 2.52% | 77.36% | 6.79 | 3.17% | 76.92% | 6.62 | 3.74% | 86.57% | **5.87** | **3.39%** | 84.78% |
| Gemini-1.5-pro | 4.37 | 2.36% | 78.62% | 4.72 | **2.78%** | 77.51% | 5.15 | 4.14% | 83.58% | 4.93 | 3.63% | 82.61% |
| Gemini-2.0-flash | 3.60 | 2.33% | 77.99% | 4.34 | 3.10% | 75.74% | 4.38 | 3.43% | 83.58% | 4.33 | 3.69% | 78.99% |
| Gemini-1.5-flash | 4.02 | **2.15%** | 77.36% | 4.37 | 2.86% | 67.46% | 4.25 | 3.48% | 80.60% | 4.38 | 3.80% | 77.54% |
| **Open-Source Models** | | | | | | | | | | | | |
| LLaMA3.1-8B | 5.23 | 2.71% | 72.96% | 6.59 | 3.50% | 74.56% | 6.53 | 3.77% | 86.57% | **5.97** | 4.11% | 82.61% |
| Qwen2.5-7B | 3.57 | 2.32% | 69.81% | 5.01 | 3.23% | 69.23% | 4.90 | 3.74% | 85.07% | 4.41 | 3.64% | 79.71% |
| BioMistral-7B | 6.18 | 3.78% | 72.96% | 6.69 | 4.66% | 65.68% | 7.24 | 4.65% | 83.58% | 6.34 | 5.60% | 77.54% |
| Phi-4 | 4.26 | 2.00% | 72.96% | 5.56 | 2.55% | **75.15%** | **5.97** | 2.98% | 82.09% | 5.85 | 3.05% | 83.33% |
| Yi-1.5-9B | 7.81 | 3.14% | 72.33% | 8.09 | 3.69% | 66.86% | 8.51 | 3.80% | **88.06%** | 8.34 | 4.55% | 82.61% |
| Mistal-7B | 6.49 | 3.12% | 75.47% | 6.94 | 3.85% | 73.37% | 6.87 | 4.04% | 85.07% | 7.38 | 3.74% | 84.06% |
| InternLM3-8B | 6.58 | 2.65% | 77.36% | 6.79 | 3.10% | 72.78% | 6.67 | 3.58% | 86.57% | 6.82 | 3.73% | 80.43% |
| Gemma-3-12b | 3.29 | **1.11%** | 73.58% | 3.75 | **1.65%** | 73.96% | 3.22 | **1.99%** | 80.60% | 3.72 | **2.25%** | 81.88% |
| DeepSeek-V2-Lite | **5.92** | 2.92% | 74.21% | 7.21 | 3.59% | 69.23% | 7.77 | 4.00% | 83.58% | 7.21 | 3.93% | 78.99% |
| LLaMA3.1-70B | 7.79 | 3.18% | **81.13%** | 8.84 | 3.81% | 74.56% | 7.83 | 4.02% | 86.57% | 7.94 | 4.50% | **84.78%** |
| Qwen-2.5-14B | 5.26 | 3.00% | 77.99% | **5.92** | 3.26% | 71.01% | 6.11 | 3.51% | **88.06%** | 5.95 | 3.92% | 81.88% |
| Qwen-2.5-32B | 2.84 | 2.38% | 61.64% | 3.64 | 3.54% | 66.27% | 4.53 | 4.33% | 77.61% | 3.60 | 4.03% | 75.36% |
| Qwen-2.5-72B | 3.13 | 2.64% | 71.07% | 3.75 | 3.50% | 72.78% | 4.31 | 3.64% | 82.09% | 3.91 | 4.14% | 78.26% |
| Target value | 6 | 0 % | 100 % | 6 | 0 % | 100 % | 6 | 0 % | 100 % | 6 | 0 % | 100 % |

series OpenAI et al. (2024a), o series OpenAI et al. (2024b), Claude Anthropic (2024), Gemini Team et al. (2024a; 2025) For open source models, we consider LLaMA 3.1 series Grattafiori et al. (2024), Qwen series Team (2024), Phi-4 Abdin et al. (2024), Mistral Jiang et al. (2024), BioMistral Labrak et al. (2024), InternLM3 Cai et al. (2024), Gemma-2 Team et al. (2024b), DeepSeek DeepSeek-AI et al. (2024), Yi-1.5 AI et al. (2025). We use their instruction versions (or chat versions) by default.

To ensure a fair comparison, when benchmarking LLMs, for proprietary models, we use their official OpenAI-compatible APIs. For open-source models, we use vLLM's openai-compatible APIs. When benchmarking post-training methods, we use Huggingface's APIs.

## 4.2 INVESTIGATION ON RQ1 AND RQ2

**Comprehensive study of readability and accuracy**  For RQ1 (Section 3.3), we explicitly set a target score, i.e., 6 for Flesch-Kincaid grade level and 0 for the professional score, and prompt it to answer the REPQA queries accordingly. Here, $\pi{=}6$ approximates a 5th grade level intended to be broadly accessible, and $\rho{=}0$ denotes the absence of domain-specific medical jargon. As shown in Table 4, most models struggle to hit the target, often overshooting complexity or, when simplifying, omitting qualifiers. Across the four categories, *definition* is the easiest, with an average accuracy of 84.06% and the lowest readability scores. More analysis of Table 4 could be found in Appendix D.

**Calibration across representative models**  We evaluate four representative LLMs: two non-thinking models (LLaMA 3.1-8B, Qwen-3-30B) and three thinking models (DeepSeek-R1-Distill-Qwen, Qwen-3-30B-Thinking, GPT-4.1), focusing on their ability to match requested reading levels. We set four target levels, i.e., 5th grade, middle school, high school, and college, corresponding to Flesch–Kincaid grade levels of 6, 9, 12, and 15. As shown in Fig. 3(a,b), all models generally follow the intended direction (Spearman $\rho \approx 0.34 \sim 0.57$) but remain poorly calibrated: their Flesch–Kincaid grades consistently fall short of the targets, with larger deviations at higher levels. In short, while models adjust readability in a relative sense, they fail in absolute alignment.

**Readability understanding of generated text**  We further test whether poor control stems from a weak readability understanding. We sample 200 generated responses per bin (Elementary, Middle School, High School, College; mapped from Flesch–Kincaid) and ask models to classify the bin. As shown in Fig. 3(c), accuracy peaks at Middle School, is moderate at Elementary, and collapses at

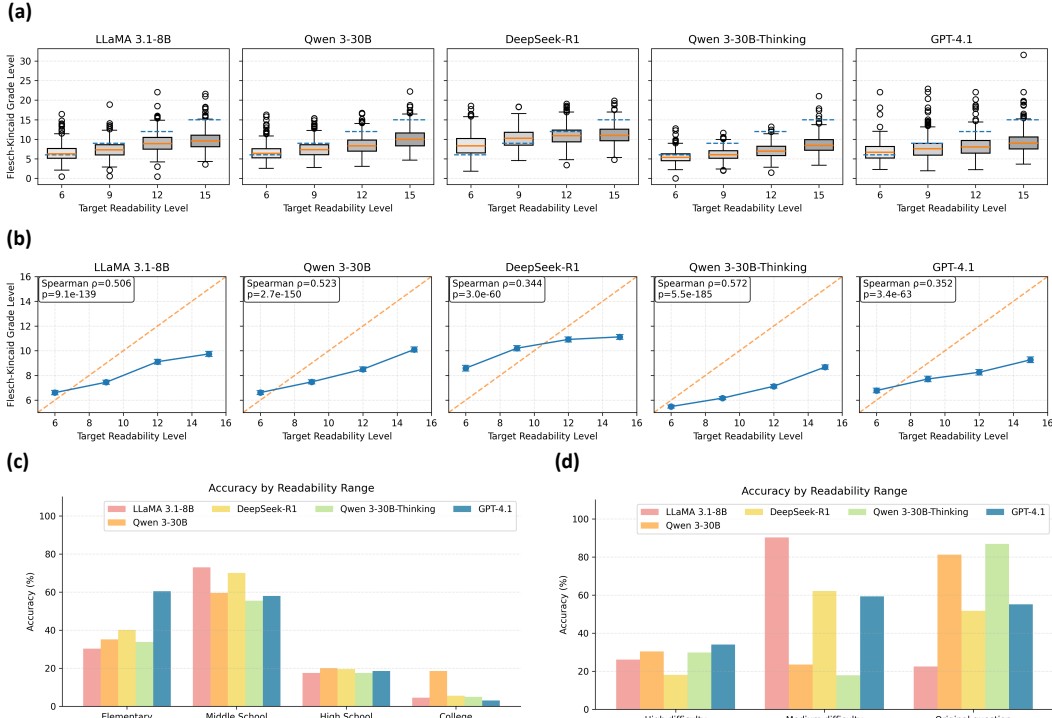

Figure 3: Readability control and understanding across four models (LLaMA 3.1–8B, Qwen-3-30B, DeepSeek-R1-Distill-Qwen, Qwen-3-30B-Thinking, GPT-4.1). **(a)** Achieved Flesch–Kincaid grades when instructed to write at target levels 6/9/12/15 (boxplots). Red horizontal lines mark the targets, and blue dashed lines mark the per-target mean grade. **(b)** Mean Flesch–Kincaid grade vs. target (error bars show 95% CI). **(c)** Accuracy of classifying generated responses into Flesch–Kincaid buckets (Elementary/Middle/High/College). **(d)** Accuracy when classifying Original/Medium/High counterparts of the same question.

High School and College. This systematic bias toward lower buckets mirrors the undershoot observed earlier, indicating that models struggle to recognize high-grade text, not only to produce it.

**Sensitivity to required medical knowledge**  For RQ2, we study whether LLMs understand differences in required medical knowledge. For each question in REPQA, we prompt GPT-4o to generate two counterparts that vary only in required domain knowledge (Medium and High). Some examples can be found in Appendix A.3. Each model then classifies the variant. As shown in Fig. 3(d), performance is uneven across systems, for example, LLaMA 3.1-8B is relatively strong on Medium but weak on Original, whereas Qwen-3-30B-Thinking excels on Original yet remains low on High. Across models, detection of High difficulty is uniformly poor, indicating limited sensitivity to advanced domain knowledge and helping explain the tendency to default to overly simple outputs.

### 4.3  RQ3: READABILITY AS REWARD SIGNAL

We optimize readability using the reward in Eq. 3 within GRPO and a token-adapted variant (TA-GRPO) that redistributes the sequence-level reward to tokens in proportion to the occurrence of professional terms. We also include standard prompting and chain-of-thought (CoT) prompting as non-RL baselines. Results are reported in Table 5.

Relative to plain prompting, CoT produces less readable text with no meaningful accuracy gain, consistent with our benchmark's emphasis on retrieval and style control rather than step-by-step reasoning. Both GRPO and TA-GRPO improve readability over the original model; TA-GRPO yields the largest gains, reducing the grade level by 41.4% on average and the professional score by 73.3%. Moreover, TA-GRPO outperforms vanilla GRPO on professional readability (53.4% average reduction) while maintaining a comparable grade level (3.0% difference). We report QA accuracy alongside these readability metrics to quantify trade-offs. Despite substantial readability gains from

Table 5: Benchmarking different readability-enhancement strategies with different backbone LLMs.

| Model | Suggestion | | | Fact | | | Definition | | | Rationale | | |
|---|---|---|---|---|---|---|---|---|---|---|---|---|
| | Grade | Prof. | Acc. | Grade | Prof. | Acc. | Grade | Prof. | Acc. | Grade | Prof. | Acc. |
| LLaMA3.1-8B | 7.64 | 4.10% | 72.33% | 8.52 | 4.69% | 71.60% | 8.74 | 5.18% | 91.04% | 8.94 | 6.74% | 76.81% |
| + Prompt | 5.23 | 2.71% | 72.96% | 6.59 | 3.50% | 74.56% | 6.53 | 3.77% | 86.57% | 5.97 | 4.11% | 82.61% |
| + CoT | 10.77 | 5.07% | 79.25% | 9.18 | 5.53% | 72.78% | 9.03 | 6.15% | 86.57% | 9.04 | 7.02% | 81.89% |
| + GRPO | 4.86 | 3.12% | 69.81% | 5.57 | 3.25% | 69.82% | 6.87 | 4.34% | 89.55% | 5.81 | 4.49% | 76.81% |
| + TA-GRPO | 5.39 | 1.37% | 64.78% | 5.89 | 1.95% | 68.04% | 5.65 | 2.23% | 85.07% | 6.05 | 2.34% | 75.36% |
| Qwen2.5-7B | 7.89 | 3.31% | 85.53% | 9.45 | 4.08% | 79.28% | 9.83 | 4.47% | 92.54% | 9.70 | 5.47% | 86.23% |
| + Lookahead | 3.20 | 1.36% | 70.45% | 3.84 | 2.00% | 68.09% | 4.31 | 2.91% | 91.67% | 3.90 | 2.49% | 81.48% |
| + Prompt | 3.57 | 2.32% | 69.81% | 5.01 | 3.23% | 69.23% | 4.90 | 3.74% | 85.07% | 4.41 | 3.64% | 79.71% |
| + CoT | 5.09 | 4.20% | 71.07% | 5.39 | 6.63% | 71.60% | 6.57 | 7.06% | 83.58% | 5.44 | 6.98% | 82.61% |
| + GRPO | 5.66 | 1.99% | 76.73% | 5.88 | 2.19% | 73.37% | 6.39 | 2.91% | 91.04% | 5.69 | 2.73% | 83.33% |
| + TA-GRPO | 5.29 | 0.90% | 75.47% | 5.53 | 0.89% | 76.33% | 6.25 | 1.74% | 91.04% | 5.05 | 0.75% | 82.61% |
| BioMistral-7B | 8.68 | 4.00% | 67.92% | 9.76 | 4.60% | 66.86% | 11.45 | 5.56% | 86.57% | 10.22 | 6.14% | 76.09% |
| + Prompt | 6.18 | 3.78% | 80.50% | 6.69 | 4.66% | 65.68% | 7.24 | 4.65% | 83.58% | 6.34 | 5.60% | 77.54% |
| + CoT | 7.53 | 4.25% | 69.81% | 7.46 | 5.13% | 68.05% | 7.73 | 5.57% | 86.57% | 6.91 | 5.86% | 71.74% |
| + GRPO | 4.53 | 1.89% | 71.07% | 4.56 | 1.89% | 68.64% | 4.69 | 2.54% | 86.57% | 4.52 | 2.40% | 75.36% |
| + TA-GRPO | 4.23 | 0.81% | 69.81% | 4.73 | 0.89% | 62.13% | 5.51 | 1.86% | 82.09% | 4.64 | 0.89% | 71.74% |
| Target value | 6 | 0 | 100 % | 6 | 0 | 100 % | 6 | 0 | 100 % | 6 | 0 | 100 % |

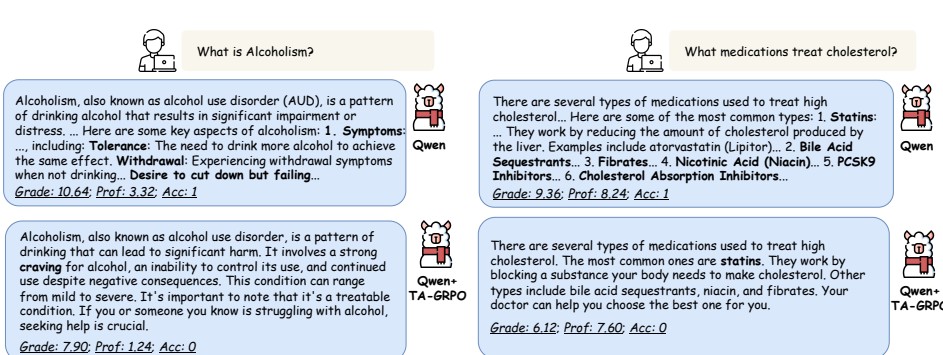

Figure 4: Two examples of improved readability but reduced accuracy.

GRPO/TA–GRPO, QA accuracy often declines, especially on *Suggestion* and *Fact*, highlighting the difficulty of simplifying without losing content.

## 4.4 CASE STUDY

Table 5 shows that, although GRPO/TA–GRPO with a readability-based reward markedly improves readability, it often reduces QA accuracy. To illustrate this trade-off, Fig. 4 presents two cases where TA–GRPO yields simpler text but misses key facts relative to the base model. In particular, the TA–GRPO model tends to over-simplify, compressing lists and collapsing categories, so only a salient term is retained (e.g., *craving* for alcohol use disorder; *statins* for cholesterol drugs) while other classes (e.g., bile-acid sequestrants, fibrates, niacin, PCSK9 inhibitors) are merged into a single undifferentiated sentence, leading to information loss.

## 5 CONCLUSION

In this study, we examine a critical factor in developing effective patient education agents, i.e., the readability of LLM-generated responses, using a collection of expert-grounded patient education QA pairs, REPQA. REPQA comprises 533 QA pairs from 27 online resources, along with a large-scale collection of patient education questions used to explore the impact of various strategies for improving readability. Our evaluation is based on two readability metrics and one accuracy metric assessed via a proxy multiple-choice QA task. We evaluate 25 LLMs on REPQA and benchmark four readability-enhancement strategies across three backbone models. The results reveal a significant limitation of current LLMs: they struggle to follow targeted readability instructions and to recognize the reading levels of both questions and text. While some post-training methods improve readability, these gains often come at the expense of accuracy. Together, these findings provide practical insights for building more accessible and trustworthy LLM-powered healthcare agents. **Limitations** Our evaluation dataset primarily targets English-language content and lay users in the U.S. healthcare context. As such, the generalizability of our findings to other languages, cultures, or health systems remains limited.

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

# A DATASET SETUP

## A.1 ONLINE PUBLIC RESOURCES

Table 6 lists all the public resources for curating high-quality question-answering pairs in the test set of REPQA. Table 7 lists all the online resources used for curating public health questions. Each link may be associated with multiple health-related categories.

Table 6: Public resources of Public Health for Curating QA Pairs.

| Topic | Links |
|---|---|
| General Health | https://www.aarp.org/health/top-health-questions/ |
| Cardiovascular | https://www.ohsu.edu/knight-cardiovascular-institute/frequently-asked-cardiovascular-questions
https://www.multicare.org/vitals/answers-to-common-heart-questions/ |
| Cholesterol | https://nyulangone.org/care-services/center-for-the-prevention-of-cardiovascular-disease/frequently-asked-questions-about-heart-health |
| Diabetes | https://diabetesaction.org/questions-general-information
https://uihc.org/health-topics/diabetes-frequently-asked-questions
https://www.nichd.nih.gov/health/topics/diabetes/more_information/other-faqs |
| HIV | https://www.unaids.org/en/frequently-asked-questions-about-hiv-and-aids |
| Diet | https://www.uclahealth.org/medical-services/weight-management/patient-resources/frequently-asked-questions
https://www.cdc.gov/bmi/faq/index.html
https://www.nutrition.gov/expert-q-a |
| Exercise | https://www.unk.edu/blog/2020/02/exercise-and-physical-activity-faqs.php |
| Alcohol | https://dmh.lacounty.gov/our-services/employment-education/education/alcohol-abuse-faq/ |
| Smoking | https://resphealth.org/quitting-smoking-faqs/ |
| Sleep | https://www.gwhospital.com/frequently-asked-questions-about-sleep
https://medlineplus.gov/ency/quiz/000805_16.htm?quiz=1
https://www.mayoclinichealthsystem.org/hometown-health/speaking-of-health/8-common-sleep-study-questions |
| Mental health | https://www.wellnessinmind.org/frequently-asked-questions/
https://www.mountsinai.org/care/psychiatry/services/depression-anxiety-disorders/faq
https://bbrfoundation.org/faq/frequently-asked-questions-about-depression
https://wmich.edu/suicideprevention/basics/faq |
| Sexual health | https://doh.wa.gov/you-and-your-family/illness-and-disease-z/sexually-transmitted-infections-sti/frequently-asked-questions
https://www.vdh.virginia.gov/adolescent-health-hub/sexual-health-faqs/ |
| Health equity | https://lgbtq.unc.edu/resources/frequently-asked-questions/
https://www.unfe.org/know-the-facts/faqs/
https://lpbcc.wordpress.com/wp-content/uploads/2012/02/ref-27-health-inequalities-q-a-final.pdf
https://www.heart.org/en/health-topics/consumer-healthcare/why-is-health-insurance-important/faqs-about-health-insurance |

Table 7: Public resources of Public Health for Curating Large Scale Healthcare Questions.

| Topics | Links |
|---|---|
| Exercise, Diet, Cardiovascular, Weight, Sleep, Cholesterol | https://www.heart.org/en/healthy-living |
| Cardiovascular | https://www.heart.org/en/about-us/heart-attack-and-stroke-symptoms |
| Cardiovascular, Cholesterol, Diabetes, Mental health, Weight | https://www.heart.org/en/health-topics |
| Exercise, Diet, Sleep, Cardiovascular, Weight, Cholesterol, Smoking | https://www.heart.org/en/healthy-living/healthy-lifestyle/lifes-essential-8 |
| Cardiovascular, Cholesterol, Diabetes | https://professional.heart.org/en/guidelines-statements |
| Others | https://www.uspreventiveservicestaskforce.org/uspstf/recommendation-topics/information-for-consumers |
| Exercise, Diet, Mental health, Smoking, Sleep | https://www.sbm.org/healthy-living |
| Cardiovascular, Sleep, Weight | https://www.nhlbi.nih.gov/health |
| Cardiovascular, Sleep, Weight | https://www.nhlbi.nih.gov/es/salud |
| Others | https://www.nhlbi.nih.gov/resources?f%5B0%5D=language%3AEnglish&f%5B1%5D=language%3ASpanish |
| Diabetes, Weight, Diet | https://www.niddk.nih.gov/health-information |
| Diabetes, Weight, Diet | https://www.niddk.nih.gov/health-information/informacion-de-la-salud |
| Diabetes, SOGI | https://www.cdc.gov/diabetes/risk-factors/diabetes-risk-lgbtq.html |
| Diabetes, SOGI | https://www.lgbtqiahealtheducation.org/wp-content/uploads/2019/07/TFIE-35_LGBT-diabetes-Brief_final2_pages.pdf |
| Cardiovascular | https://www.ahajournals.org/doi/10.1161/CIR.0000000000001209 |
| Cardiovascular | https://www.ahajournals.org/doi/full/10.1161/CIR.0000000000001003 |
| Mental health | https://psycnet.apa.org/fulltext/2024-79040-001.html |
| SOGI | https://translegislation.com/ |
| SOGI | https://www.aclu.org/legislative-attacks-on-lgbtq-rights-2024 |
| SOGI | https://medlineplus.gov/lgbtqiahealth.html |
| SOGI | https://lgbtqhealthcaredirectory.org/ |
| SOGI | https://www.lgbtqiahealtheducation.org/resources/ |
| SOGI | https://www.hrc.org/resources |
| Mental health | https://www.samhsa.gov/find-help |
| Mental health | https://www.apa.org/topics |
| Mental health | https://www.nimh.nih.gov/health |
| Smoking | https://smokefree.gov/ |
| Cardiovascular, Smoking | https://www.ahajournals.org/doi/10.1161/CIR.0000000000000625 |
| HIV care | https://clinicalinfo.hiv.gov/en/guidelines |
| HIV care | https://www.hiv.gov/hiv-basics |
| HIV care | https://www.hivguidelines.org/ |
| HIV care | https://www.cdc.gov/hivpartners/php/index.html |
| HIV care | https://www.who.int/news-room/fact-sheets/detail/hiv-aids |
| Sleep | https://www.nhlbi.nih.gov/health/heart-healthy-living/sleep |
| Others | https://www.drugs.com/ |
| Others | https://www.ncbi.nlm.nih.gov/books/NBK430685/ |
| Diabetes | https://pro.aace.com/clinical-guidance/diabetes |
| Diabetes | https://professional.diabetes.org/standards-of-care |
| Others | https://goldcopd.org/2024-gold-report/ |
| Others | https://www.cdc.gov/vaccines/hcp/imz-schedules/index.html |
| Others | https://www.cancer.org/cancer/types/colon-rectal-cancer/detection-diagnosis-staging/acs-recommendations.html |
| Others | https://www.asccp.org/clinical-practice-guidelines |
| Others | https://www.acog.org/clinical/clinical-guidance/practice-bulletin/articles/2017/07/breast-cancer-risk-assessment-and-screening-in-average-risk-women |
| Others | https://www.asam.org/quality-care/clinical-guidelines/alcohol-withdrawal-management-guideline |
| Others | https://www.asam.org/asam-criteria/implementation-tools/criteria-intake-assessment-form |
| Others | https://readable.com/readability/flesch-reading-ease-flesch-kincaid-grade-level/ |
| Others | https://www.spanishreadability.com/fernandez-huerta-readability-index |
| Others | https://www.mccc.edu/nursing/documents/NRS225TherapeuticCommunications.pdf |
| Others | https://www.reddit.com/r/hivaids/?share_id=aoMnalJwXK4SBcVZtxc40&utm_content=1&utm_medium=ios_app&utm_name=ioscss&utm_source=share&utm_term=4 |
| HIV care | https://forums.poz.com/index.php?PHPSESSID=qe2trb4da2kmn0dch13402q9u2&action=forum |
| HIV care | https://h-i-v.net/forums |

## A.2 STRATEGIES OF CURATING DATASET

To curate a high-quality set of patient education question-answer (QA) pairs, the initial content collection was conducted manually by annotators, most of whom are PhD students (Table 8). They identified and extracted QA-relevant content from reputable public health websites, as listed in Table 6. The annotators focused on capturing both explicit QA formats (e.g., FAQ sections) and implicit pairs derived from health articles, clinical guidelines, and symptom descriptions. The extracted content was subsequently organized and labeled by domain experts in nursing and public health to ensure accuracy and domain relevance.

To generate diverse and high-quality healthcare problems, annotators systematically reviewed all primary links and their subpages across each public health website listed in Table 7. The resources

examined included articles, PDF documents, clinical guidelines, and other educational materials. After collecting the relevant content, it was segmented into coherent text chunks based on topical boundaries and structural cues. These chunks were then used as input to an LLM (GPT-4o mini, in our case), which generated multiple healthcare problems designed to be diverse and user-oriented, guided by specific prompt instructions. Each generated problem was subsequently reviewed and validated by domain experts in nursing and public health to ensure factual accuracy, clinical relevance, and clarity.

Table 8: Biographies of all annotators involved in REPQA construction.

| ID | Year | Major | Assigned Task | Expert Validator? |
|----|------|-------|---------------|-------------------|
| 1 | 3rd year PhD | Computer Science | Scrape Website Content | ✗ |
| 2 | 3rd year PhD | Computer Science | Scrape Website Content | ✗ |
| 3 | 3rd year PhD | Computer Science | Scrape Website Content | ✗ |
| 4 | 4th year PhD | Electrical Engineering | Scrape Website Content | ✗ |
| 5 | 2nd year Master | Computer Science | Scrape Website Content | ✗ |
| 6 | 4th year PhD | Data Science | Scrape Website Content | ✗ |
| 7 | 4th year PhD | Nursing & Public Health | Collect Dataset | ✓ |
| 8 | 2nd year Master | Public Health | Collect Dataset | ✓ |
| 9 | 1st year PhD | Nursing & Public Health | Collect Dataset | ✓ |
| 10 | 3st year PhD | Nursing & Public Health | Collect Dataset | ✓ |

## A.3 EXAMPLES OF GENERATED QUESTIONS WITH HIGH AND MEDIUM DIFFICULTIES

In Section 3.3 (RQ2), we extend the original questions into two counterparts of medium and high difficulty using the prompt provided in Appendix B.2.2, and apply them for classification. Below, we present several examples of the generated questions:

---

**Example 1**

"original_question": "What are the negative effects of smoking?"
"medium_difficulty": "How does smoking affect cardiovascular health?"
"high_difficulty": "What are the carcinogenic mechanisms of tobacco?"

---

**Example 2**

"original_question": "Is vaping actually bad for you?"
"medium_difficulty": "What are the long-term health risks of vaping?"
"high_difficulty": "Discuss the pathophysiology of e-cigarette lung injury."

---

**Example 3**

"original_question": "How many calories should I eat in a day?"
"medium_difficulty": "How is daily caloric need calculated?"
"high_difficulty": "What is the Harris-Benedict equation for BMR?"

---

**Example 4**

"original_question": "What are the types of diabetes?"
"medium_difficulty": "Distinguish between Type 1 and Type 2 diabetes."
"high_difficulty": "Explain the autoimmune pathology of Type 1 diabetes."

---

**Example 5**

"original_question": "Can I smoke with high blood pressure?"
"medium_difficulty": "How does smoking affect hypertension management?"
"high_difficulty": "Explain nicotine's effect on arterial blood pressure."

---

## A.4  PROXY MULTIPLE CHOICE QA GENERATION

To quantify the accuracy of the generated responses by LLMs, we introduce a proxy multiple-choice QA task. The multi-choice QA pairs are generated based on the curated QA pairs, following the Algorithm 1.

---

**Algorithm 1:** Proxy Multiple Choice QA Generation

---

**Input** : Public health QA pairs $\mathcal{D}$
**Output:** Multiple choice QA sets $\mathcal{D}^{\mathrm{mc}} = \{d_i^{\mathrm{mc}}\}$, where $d_i^{\mathrm{mc}}$ is a triplet $(a_i, q_i^{\mathrm{mc}}, a_i^{\mathrm{mc}})$ with $a_i$ the reference answer, $q_i^{\mathrm{mc}}$ the generated multiple-choice question, and $a_i^{\mathrm{mc}}$ the correct option
**Function** `Generate(`*q, a*`)`:

    `// Proposer step`
    $q^{\mathrm{mc}}, a^{\mathrm{mc}} \leftarrow$ `LLM-Proposer(`*q, a*`)`
    `// Step 1:  Sanity check with LLM critic on reference`
    $s^+ \leftarrow$ `LLM-Critic(`$q^{mc}, a$`)`
    `// Step 2:  Sanity check with LLM critic on irrelevant`
    `   context`
    $s^- \leftarrow$ `LLM-Critic(`$q^{mc}, a_{noise}$`)`
    **if** $s^+ \wedge \neg s^-$ **then**
        `// Provide feedback to proposer and regenerate`
        prompt $\leftarrow$ updated prompt
        **return** `Generate(`*q, a*`)`
    **end**
    **return** $q^{\mathrm{mc}}, a^{\mathrm{mc}}$
`// Main procedures to generate all multiple-choice QAs`
$\mathcal{D}^{\mathrm{mc}} \leftarrow \emptyset$
prompt $\leftarrow$ default prompt
**foreach** *QA pair* $(q_i, a_i) \in \mathcal{D}$ **do**
    $q_i^{\mathrm{mc}}, a_i^{\mathrm{mc}} \leftarrow$ `Generate(`$q_i, a_i$`)`
    $\mathcal{D}^{\mathrm{mc}} \leftarrow \mathcal{D}^{\mathrm{mc}} \cup \{(a_i, q_i^{\mathrm{mc}}, a_i^{\mathrm{mc}})\}$
**end**

---

# B  EXPERIMENTAL SETUP

## B.1  CONFIGURATION OF MODELS

We set the learning rate to $1 \times 10^{-5}$ and apply early stopping, using a validation set split from the training data. For post-training, we adopt LoRA Hu et al. (2021) with a rank of 16. In the reward formulation (Equ.4), we set the coefficients as $\alpha = 1$, $\beta = 1$, and $\gamma = 50$. For TA-GRPO (Equ.5), we use a weighting factor of $w = 2$. Each training run is performed on an A40 GPU for 48 hours.

During training with GRPO and TA-GRPO, we use a temperature of $\tau = 0.9$ and set the maximum generation length to 256 tokens. For inference during evaluation, we adopt greedy decoding with a maximum length of 2048 tokens. All other generation hyperparameters follow the default settings in `https://huggingface.co/docs/transformers/en/main_classes/text_generation#transformers.GenerationConfig`.

## B.2    PROMPT FOR LLMs

### B.2.1    PROMPT FOR PROXY MULTIPLE CHOICE QA

---

**System Prompt for LLM Proposer**

You are a helpful assistant. Given the reference, which is a QA pair, your task is to create a multiple-choice question with one correct answer and three distractors based on the reference QA. Write the question as if it stands alone, without any indication that it was derived from a specific reference. For example, phrases like "according to the reference" are not allowed in your created question. The distractors should be as plausible as possible but either 1) incorrect or 2) correct but cannot be inferred from the reference. Format your output exactly as:
Q: <question>
A. <correct_answer>
B. <distractor1>
C. <distractor2>
... (possibly other options)
Solution: <the letter of the correct answer>. Do not include any text that is not part of the question in any round of the conversation. Your question should be as difficult as possible. Distractors that are obviously wrong should not be considered.

---

**System Prompt for LLM Critic**

You are a helpful assistant. Given a multiple-choice question and a reference document, your task is to give the correct answer letter based on the document. The format of the output should exactly be: "the letter of the correct answer", i.e., output the correct letter directly. For example, if the correct answer is A, you should output "A". Output "I do not know" if you cannot find the answer based on the document given. Do not include any other text or explanation in your response. Do not include any intermediate steps or reasoning. Do not use your own knowledge other than what is given in the document to answer the question. That is to say, you should output "I do not know" when you cannot infer the solution from the document given, even if you know the solution.

---

### B.2.2    PROMPT FOR RQ1 AND RQ2

---

**System Prompt for Instruction Following**

You are a helpful agent in answering questions related to health. Answer clearly and kindly at a {5th grade/middle school/high school/college} reading level (U.S. grade). Maintain a positive sentiment and conversational tone.

---

---

**System Prompt for Reading Level Inference**

You are a careful readability assessor for healthcare and science texts. Your job is to classify the readability/audience level of a given paragraph.

Definitions (pick exactly one):
1) Elementary school: very simple words and ideas; no jargon; everyday concepts only.
2) Middle school: mostly common words; some domain terms with brief explanations; moderately short sentences; basic causal reasoning.
3) High school: frequent domain terms and abstractions; complex sentences and clauses; assumes some prior science knowledge; limited definitions of jargon.
4) College: specialized terminology or references; dense concepts and implicit assumptions; long/complex sentences; minimal or no lay explanations.

Instructions:
1) Judge solely from the paragraph provided. Do not invent context.
2) Choose ONE label from: "Elementary school","Middle school","High school","College".

Output format:
1) You may include optional analysis.
2) The LAST line must be exactly: 'Final Answer: <ONE WORD from "Elementary school","Middle school","High school","College">'
3) No extra words after the label on that last line.

---

**System Prompt for Generating High and Medium Levels Counterpart of Original Questions**

You are an expert in public health communication and medical education. Your task is to transform a simple public health question, aimed at the general public, into two more challenging versions: one "medium-difficulty" version and one "high-difficulty" version.

Category Definitions:
1. Medium Difficulty: This question should be something an educated and curious patient might ask their doctor. It is more specific than the original, may use some common medical terms, but remains generally accessible.
2. High Difficulty: This question should be one that a medical student, clinician, or researcher would ask. It must be highly technical, use specific jargon or acronyms, and assume significant background knowledge.

Core Constraints:
1. Length: The two new questions you generate MUST be of a similar length to the original question. Focus on increasing the question's conceptual density, not just its word count.
2. Output Format: Your output must be in a strict JSON format, containing three keys: original_question, medium_difficulty, and high_difficulty.

Examples:

Input Question 1:
"How can I quit smoking?"

Required JSON Output:
{
"original_question": "How can I quit smoking?",
"medium_difficulty": "What are the best smoking cessation aids?",
"high_difficulty": "What are the contraindications for varenicline therapy?"
}

---

---

**System Prompt for Predicting Needed Readability Level for a Question**

You are an expert in public health communication and text analysis. Your task is to classify a given public health question based on the conceptual complexity and the level of readability required to provide a comprehensive answer.
There are three categories:
1. Simple: The question is for the general public, uses common language, and asks about general knowledge. An answer would require a low Flesch-Kincaid Grade Level.
2. Intermediate: The question is for a curious and educated layperson. It is more specific, may use some common medical terms, and requires a more detailed answer than a simple question.
3. Complex: The question is for a medical student, healthcare professional, or researcher. It is highly technical, uses specific medical jargon or acronyms, and assumes significant background knowledge. An answer would require a high Flesch-Kincaid Grade Level.
Read the question below and classify it into one of the three categories. Respond with only one word: 'Simple', 'Intermediate', or 'Complex'.

### B.2.3 PROMPT FOR BENCHMARKING READABILITY-ENHANCEMENT STRATEGIES

**Standard System Prompt**

You are a helpful agent in answering questions related to health. Please respond to questions based on credible information and use simple terms (limit medical jargon and 5th grade reading level). Answers should be short ($\sim$63 tokens) with simple sentence structure, and words should be easy to understand (fewer syllables). Maintain a positive sentiment and conversational tone.

**CoT System Prompt**

You are a helpful agent in answering questions related to health. Please respond to questions based on credible information and use simple terms (limit medical jargon and 5th grade reading level). The final answers (excluding reasoning steps) should be short ($\sim$63 tokens) with simple sentence structure, and words should be easy to understand (fewer syllables). Begin by thinking through the problem step by step, clearly showing your reasoning. After completing your reasoning, provide your final conclusion, exactly starting with 'Final Answer:'.

**System Prompt for avoiding trivial solution in GRPO and TA-GRPO**

Given a response to a question, you are required to judge if it is a useful answer. For example, responses like "I do not have enough information" and "I can provide general information regarding the topic" (but actually no useful information at all) are not useful answers because they give no solution to the question. Some answers that just rephrase the question without giving specific extra information should be considered as not useful as well. Answer yes if it is useful or no otherwise.

## C  READABILITY METRICS

### C.1  FLESCH-KINCAID GRADE LEVEL

Table 9 presents the correspondence between the Flesch-Kincaid Grade Level scores and U.S. school grade levels.[3] We use the `readability` Python library[4] to compute this metric and assess the readability of generated content.

---

[3] https://readable.com/readability/flesch-reading-ease-flesch-kincaid-grade-level/
[4] https://pypi.org/project/readability/

Table 9: The relationship between Flesch-Kincaid grade level and grade level.

| Flesch-Kincaid Score | Reading Level | School Level | Age Range (US) |
|---|---|---|---|
| 0 - 3 | Basic | Kindergarten / Elementary | 5 - 8 |
| 3 - 6 | Basic | Elementary | 8 - 11 |
| 6 - 9 | Average | Middle School | 11 - 14 |
| 9 - 12 | Average | High School | 14 - 17 |
| 12 - 15 | Advanced | College | 17 - 20 |
| 15 - 18 | Advanced | Post-grad | 20+ |

## C.2 TOKEN-ADAPTED GRPO

The standard GRPO operates at the sentence level, assigning a uniform reward across all the tokens within the generated responses. However, this uniform distribution mitigates the influence of medical jargon on the readability metrics, i.e., the professional score, which depends on the presence of domain-specific terms at certain specific tokens.

In light of this, we propose a token-adapted variant of GRPO that applies a weighted distribution of rewards across tokens. Specifically, at each optimization step, the LLM generates a group of responses $\{o_1, \cdots, o_G\}$ where $o_i$ denotes $i$-th generated response. For $t$-th token in $i$-th response, a token-level reward is computed based on whether it appears in our curated professional vocabulary:

$$r'_{i,t} = \frac{w_{i,t}T}{\sum_{t'}^{T} w_{i,t'}} r_i$$

$$\text{where} \quad w_{i,t} = 1 + \delta(o_{i,t})w \tag{4}$$

where $r_i$ is the original sentence-level reward for the response $o_i$, $T$ is the maximum length of the responses, and $\delta(o_{i,t})$ is an indicator function that returns 1 if the token $o_{i,t}$ belongs to a professional term, and 0 otherwise. The hyperparameter $w$ controls the penalty strength for professional terms. Using these token-level rewards, we redefine the advantage function in GRPO as:

$$A_{i,t} = \frac{r'_{i,t} - \text{mean}(\{r_1, \cdots, r_G\})}{\text{std}(\{r_1, \cdots, r_G\})} \tag{5}$$

In our implementation, we apply this token-level reward to the professional score while maintaining the original sentence-level reward formulation for the Flesch-Kincaid grade level metric.

## C.3 PROFESSIONAL SCORE

We construct the vocabulary by collecting professional terms from the Harvard Public Health medical dictionary.[5] Specifically, annotators scraped terminology entries from the site, after which a subset of terms was removed based on a review by public health experts. The vocabulary is available at `https://anonymous.4open.science/r/RepQA-45A8/dataset/harvard_medical_terms.json`.

## D MORE ANALYSIS

Among these models, `Claude-3.7-Sonnet` outputs responses with the highest average grade level, while `Qwen2.5-32B` achieves the lowest grade level, and `LLaMA3.1-8B` is the closest to our target value (i.e., 6). As for the professional score, `BioMistral-7B` produces the most specialized language, likely due to its fine-tuning on medical corpora. In contrast, `Gemma3-12b` yields the lowest professional word usage, which is preferable for readability. As for accuracy, `o4-mini` attains the highest score at 83.68%.

Across the four evaluated categories (*suggestion*, *fact*, *definition*, *rationale*), the *definition* category is the easiest, with an average accuracy of 84.06%, whereas the *fact* category is the most challenging, with an average accuracy of 72.80%. Notably, the *suggestion* category consistently exhibits both the

---

[5]`https://www.health.harvard.edu/a-through-c`

Table 10: Benchmarking both proprietary and open-source LLMs.

| Model | Flesch-Kincaid | Dale Chall | SMOG | Yngve Depth |
|---|---|---|---|---|
| **Proprietary Models** | | | | |
| o4-mini | 3.79 | 7.03 | 5.53 | 2.84 |
| o3-mini | 3.99 | 6.72 | 5.56 | 2.55 |
| GPT-4o | 5.00 | 7.56 | 6.97 | 2.63 |
| GPT-4o-mini | 4.42 | 7.07 | 6.62 | 2.56 |
| GPT-4.1 | 4.31 | 6.74 | 5.93 | 2.68 |
| GPT-4 | 4.46 | 6.89 | 6.28 | 2.57 |
| GPT-4-turbo | 5.18 | 6.69 | 5.97 | 2.66 |
| Claude-3.7-Sonnet | 8.63 | 8.43 | 7.60 | 3.60 |
| Claude-3.5-Sonnet | 6.50 | 7.59 | 6.44 | 2.99 |
| Gemini-1.5-pro | 4.73 | 6.87 | 6.82 | 2.63 |
| Gemini-2.0-flash | 4.12 | 6.45 | 6.06 | 2.53 |
| Gemini-1.5-flash | 4.25 | 7.10 | 6.47 | 2.44 |
| **Open-Source Models** | | | | |
| LLaMA3.1-8B | 6.27 | 8.22 | 8.19 | 2.89 |
| Qwen2.5-7B | 4.49 | 6.59 | 5.72 | 2.57 |
| Phi-4 | 5.30 | 7.21 | 7.49 | 2.76 |
| Yi-1.5-9B | 8.12 | 8.56 | 8.96 | 2.98 |
| Mistal-7B | 6.91 | 9.22 | 7.55 | 2.84 |
| InternLM3-8B | 6.72 | 8.31 | 7.60 | 3.05 |
| Gemma-3-12b | 3.54 | 7.87 | 6.69 | 2.49 |
| DeepSeek-V2-Lite | 6.89 | 8.29 | 8.30 | 2.82 |
| LLaMA3.1-70B | 8.17 | 8.78 | 8.63 | 3.21 |
| Qwen-Qwen2.5-14B | 5.76 | 7.35 | 6.55 | 2.73 |
| Qwen-Qwen2.5-32B | 3.50 | 6.34 | 5.03 | 2.41 |
| Qwen-Qwen2.5-72B | 3.68 | 6.73 | 5.05 | 2.49 |

lowest Flesch-Kincaid grade level and professional score, which aligns with intuition: suggestions typically do not require complex language compared to the explanation of medical facts or rationales.

Among proprietary models, we observe that more advanced versions within a series consistently demonstrate improved accuracy. For instance, accuracy increases with model upgrades such as `o3-mini` → `o4-mini` and `GPT-4-turbo` → `GPT-4` → `GPT-4.1`. These trends indicate that newer model iterations are more effective in generating accurate responses.

## E   MORE READABILITY METRICS

Table 10 reports additional readability scores. Alongside the Flesch–Kincaid score, we include Dale–Chall Tanprasert & Kauchak (2021), SMOG Mc Laughlin (1969), and Yngve Depth Yngve (1960). These metrics are all strongly correlated with Flesch–Kincaid (Dale–Chall: 0.800; SMOG: 0.806; Yngve Depth: 0.823), the primary readability measure used in the main text. The close agreement across metrics further supports the robustness of our conclusions.

## F   LLM USAGE

Large language models were used only for editorial polishing (grammar, style, and readability). They did not generate content or conduct analyses. All changes were reviewed by the authors.

