# OpenReview forum: "RepQA: Evaluating Readability of Large Language Models in Patient Education Question Answering"
_ICLR.cc/2026/Conference — ICLR 2026 Conference Desk Rejected Submission_

### Official Review · Reviewer_nYWC · 2025-11-01

**Soundness:** 3
**Presentation:** 2
**Contribution:** 2
**Rating:** 4
**Confidence:** 4

**Summary:**

The paper introduces REPQA, a benchmark for measuring how well patient-education QA system adapt to readable language. It includes 533 expert-valudated QA pairs across 4 intent types such as Suggestions, Facts, Definitions and Rationales with a different target reading levels. The benchmark uses Flesch-Kincaid Grade Level and professional-term ratio to assess readability as well as provides about 36k unlabelled questions for post-training. The authors evaluate 25 LLMs and three tasks (grade-level instruction following, reading-level inference and readability-aware optimisation using GRPO and a proposed Token-Adapted GRPO). The results seem to indicate that LLMs overshoot the requested grade level; TA-GRPO improves readability but harms accuracy.

**Strengths:**

* A well focused benchmark for patient health literacy. REPQA address a real deployment need of producing patient-friendly answers at 6th to 8th grade levels (recommended by US health agencies). It seems to fill a gap between professional-exam benchamarks and consumer-question datasets (e.g. HealthSearchQA within MultiMedQA).

* Practical proxy for answer quality. The propose-critic (LLM-as-judge) is a reasonable and low cost approximation to human eval of answer sufficiency, consistent with recent trends in automatic LLM evalution.

* Broad empirical study across 25 models. The paper rather well documents a readability-accuracy trade-off across by testing 25 models and shows TA-GRPO achieves the biggest improvements in grade level and  jargon reductions. This is consistent with patterns in readability-controlled generation.

* Useful insights for practitioners. The study highlights that instruction-following for readability is poorly calibrated but CoT improves answer quality without significantly hurting reading level, which seems to be also supported by evidence from prior work.

**Weaknesses:**

* Lack of comparison to establish readability-control methods. Prior work showed strong readability control (instruction tuning, other RL methods and decoding strategies) and readability-controlled medical generation [1-4]. A direct comparison against the baselines woudl help to clarify further the contribution.

* Limited readability measurement. The paper relies on FKGL, which can be "gamed" and may not capture true language difficulty. Patient-education research typically reports SMOG and Dale Chall [5,6], which are not reported in the current work.

* Unvalidated LLM-as-judge setup. LLM judge may suffer from self-preference and positional biases. Without a human-scored calibration subset to estimate agreement and CI, changes attributed to TA-GRPO remain quite uncertain.

* Scope and generalisability. The labelled test set seems somewhat limited (n=533), English-only and US-centric. This limits claims about multilingual or cross-population applicability.

[1] Ribeiro, L. F. R., Bansal, M., & Dreyer, M. (2023). Generating summaries with controllable readability levels. EMNLP. https://aclanthology.org/2023.emnlp-main.714.pdf
[2] Luo, Z., Xie, Q., & Ananiadou, S. (2022). Readability controllable biomedical document summarization.
EMNLP. https://aclanthology.org/2022.findings-emnlp.343/
[3] Wang, P., Chen, L., Zhu, D., Liu, Q., & Liu, L. (2024). Large Language Models are not Fair Evaluators. ACL. https://aclanthology.org/2024.acl-long.511.pdf
[4] Ji, Y., Li, Z., Meng, R., Sivarajkumar, S., Wang, Y., Yu, Z., … He, D. (2024). RAG-RLRC-LaySum at
BioLaySumm: Integrating retrieval-augmented generation and readability control for layman summarization of biomedical texts. BioNLP @ ACL. https://aclanthology.org/2024.bionlp-1.75/
[5] Tanprasert, T., Kauchak, D., et al. (2021). Flesch-Kincaid is not a text simplification evaluation metric. GEM Workshop. https://aclanthology.org/2021.gem-1.1/
[6] McLaughlin, G. H. (1969). SMOG grading—A new readability formula. Journal of Reading, 12(8), 639–646. Journal of Reading. https://ogg.osu.edu/media/documents/health_lit/WRRSMOG_Readability_Formula_G.Harry_McLaughlin__1969.pdf

**Questions:**

1. Could you please report results using additional readability metrics (previously mentioned, such as SMOG, Dale–Chall), or explain why these were not included? If this is possible, would you commit to including them in the final version of the paper to strengthen construct validity?

2. LLM-based evaluation can display self-preference and positional bias, did you run any calibration study with human raters to validate the LLM-judge outputs? If not, could you include a small human-judged subset (agreement + CI) to verify accuracy trends under TA-GRPO?

3. The dictionary-based approach may have limited recall, especially for multi-word medical terms. Could you clarify the coverage of your terminology list and whether multi-word medical terms are captured?

---

> ### Author Response · Authors · 2025-11-22
>
> ```
> C1: The author should include other readability-control generation baselines.
> ```
> We thank the reviewer for the helpful suggestion. Following your recommendation, we have added the Lookahead [1] decoding method as an additional readability-control baseline in Table 5. Our experiments show that Lookahead does not yield satisfactory performance on REPQA: while it slightly improves readability, it comes at the cost of noticeably lower QA accuracy, and the resulting responses still do not achieve the desired readability levels, which again highlights the significance of our benchmark. We plan to include other baselines in future iterations.
>
> ```
> C2: The author should include SMOG and Dale Chall.
> ```
> Thanks for the suggestion. We have included these two metrics, as well as the Yngve depth, which is used to measure syntax complexity.
>
> | Model | Grade | Dale Chall | Smog | Yngve |
> | --- | --- | --- | --- | --- |
> | o4-mini | 3.79 | 7.03 | 5.53 | 2.84 |
> | o3-mini | 3.99 | 6.72 | 5.56 | 2.55 |
> | GPT-4o | 5.00 | 7.56 | 6.97 | 2.63 |
> | GPT-4o-mini | 4.42 | 7.07 | 6.62 | 2.56 |
> | GPT-4.1 | 4.31 | 6.74 | 5.93 | 2.68 |
> | GPT-4 | 4.46 | 6.89 | 6.28 | 2.57 |
> | GP-4-turbo | 5.18 | 6.69 | 5.97 | 2.66 |
> | Claude-3.7-sonnet | 8.63 | 8.43 | 7.60 | 3.60 |
> | Claude-3.5-sonnet | 6.50 | 7.59 | 6.44 | 2.99 |
> | Gemini-1.5-pro | 4.73 | 6.87 | 6.82 | 2.63 |
> | Gemini-2.0-flash | 4.12 | 6.45 | 6.06 | 2.53 |
> | Gemini-1.5-flash | 4.25 | 7.10 | 6.47 | 2.44 |
> | LLaMA-3.1-8B | 6.27 | 8.22 | 8.19 | 2.89 |
> | Qwen2.5-7B | 4.49 | 6.59 | 5.72 | 2.57 |
> | Phi-4 | 5.30 | 7.21 | 7.49 | 2.76 |
> | Yi-1.5-9B | 8.12 | 8.56 | 8.96 | 2.98 |
> | Mistral-7B | 6.91 | 9.22 | 7.55 | 2.84 |
> | InternLM-8B | 6.72 | 8.31 | 7.60 | 3.05 |
> | Gemma-3-12b | 3.54 | 7.87 | 6.69 | 2.49 |
> | DeepSeek-V2-Lite | 6.89 | 8.29 | 8.30 | 2.82 |
> | LLaMA-3.1-70B | 8.17 | 8.78 | 8.63 | 3.21 |
> | Qwen2.5-14B | 5.76 | 7.35 | 6.55 | 2.73 |
> | Qwen2.5-32B | 3.50 | 6.34 | 5.03 | 2.41 |
> | Qwen2.5-72B | 3.68 | 6.73 | 5.05 | 2.49 |
>
> These metrics are all strongly correlated with Flesch–Kincaid (Dale–Chall: 0.800; SMOG: 0.806; Yngve Depth: 0.823), the primary readability measure used in the main text. The close agreement across metrics further supports the robustness of our conclusions.
>
> ```
> C3: The author should include human evaluation.
> ```
> Our dataset construction and experimental design were supported by an interdisciplinary team with nursing and public health expertise (as noted in Table 8), ensuring the methodological soundness of our study. We acknowledge that beyond in-silico evaluation, human evaluation is essential. Due to the limited rebuttal period, we cannot complete such evaluations at this stage, but we intend to incorporate a thorough human-subject assessment in our subsequent manuscript.
>
> ```
> C4: The dataset is English-only and US-centric. This limits claims about multilingual or cross-population applicability.
> ```
> We appreciate the reviewer’s observation. This has been explicitly included in the limitation section of the original manuscript. While the current dataset is English-only and U.S.-centric, we view extending REPQA to additional languages and populations as a valuable direction for future work.
>
> ```
> Q1: The author should include SMOG and Dale-Chall.
> ```
> Has been solved in C2.
>
> ```
> Q2: The author should include human evaluation and report agreement + CI.
> ```
> Has been solved in C2.
>
> ```
> Q3: Given that the dictionary-based approach may be tricky to implement, do authors consider multi-word medical terms? And what is the terminology list?
> ```
> The medical terms are from https://www.health.harvard.edu/a-through-c. Our implementation explicitly handles multi-word medical terms as well as different morphological variants of the same term (e.g., inflame → inflammation). The matching logic was designed to capture all corner cases, and we have thoroughly tested the implementation.
>
> [1] Ribeiro, L. F. R., Bansal, M., & Dreyer, M. (2023). Generating summaries with controllable readability levels. EMNLP.

---

### Official Review · Reviewer_aBjM · 2025-11-04

**Soundness:** 3
**Presentation:** 3
**Contribution:** 1
**Rating:** 4
**Confidence:** 4

**Summary:**

The paper introduces a QA benchmark aimed at investigating, the readability of LLM-generated healthcare related questions. The questions are sourced from different benchmarks, categorised  filtered by what the paper refers to as experts, and evaluated with regards to readability metrics, (proxy) accuracy scores and the density of jargon. Results indicate that across the board, accuracy is high, jargon is ostensibly low, and readability as measured by the FK index is variable. The paper also proposes to explicitly incorporate the FK score as training feedback into the traiing process (unsurprisingly) improves the metric, but reduces the accuracy.

**Strengths:**

The paper is well written, the experiments seem to be sound, the RQs are clear and the findings in general accessible and in line with the RQs.

**Weaknesses:**

The magnitude of the contribution does not feel appropriate with the target conference: This is a benchmark that does not introduce new data or methods (bar RL post-training on a metric, but this is hardly novel), and the findings are not very profound either. I have doubts whether the selected readability indices indeed reflect human judgement on the readability of the texts.
While the data is curated by "experts", the target - i.e. lay readers - are not considered in this paper.

Incorporating human feedback and correlating model performance with human feedback & automated metrics would be more interesting - it would also give more insights whether training on automated, easy to obtain metrics (e.g. FK index) would improve the human perception of readability or vice versa.

**Questions:**

Can you include human experiments into the empirical study.

---

> ### Author Response · Authors · 2025-11-22
>
> ```
> C1: The magnitude of the contribution does not feel appropriate with the target conference: This is a benchmark that does not introduce new data or methods, and the findings are not very profound either. There are doubts whether the selected readability indices indeed reflect human judgment on the readability of the texts.
> ```
> We would like to politely point out that our work indeed introduces new datasets. As stated in section 3.1, we have constructed a large-scale dataset for post-training (line 238), a high-quality, expert-reviewed dataset for testing (line 198), as well as an evaluation protocol for accuracy (line 219). While methodological novelty is not our main focus.
>
> ```
> C2: While the data is curated by "experts", the target - i.e., lay readers - are not considered in this paper.
> Incorporating human feedback and correlating model performance with human feedback & automated metrics would be more interesting - it would also give more insights whether training on automated, easy to obtain metrics (e.g., FK index) would improve the human perception of readability or vice versa.
> ```
> Our dataset construction and experimental design were supported by an interdisciplinary team with nursing and public health expertise (as noted in Table 8), ensuring the methodological soundness of our study. We acknowledge, that beyond in-silico evaluation, human evaluation for both accuracy and readability are essential, and we plan to incorporate it comprehensively in our next manuscript.
>
> ```
> Q1: Can the author include human experiments into the empirical study.
> ```
> Answered in C1.

---

### Official Review · Reviewer_Df4s · 2025-11-10

**Soundness:** 3
**Presentation:** 2
**Contribution:** 3
**Rating:** 4
**Confidence:** 4

**Summary:**

The paper introduces REPQA, a benchmark for evaluating the readability of LLM-generated responses in the patient education domain. The dataset consists of 533 expert-reviewed QA pairs drawn from 27 public health sources, covering common topics such as sleep, diet, mental health, and chronic diseases. The models are evaluated on two readability metrics (Flesch–Kincaid grade level and professional term ratio) and a proxy multiple-choice QA task for factual accuracy.
The authors also propose a token-adapted GRPO method which  improves readability metrics substantially but often at the cost of factual precision.

**Strengths:**

1. Novel and Relevant Research Focus: The paper tackles a crucial and under explored problem in healthcare AI. While prior research has primarily emphasized on factual accuracy and reasoning, this study shifts the focus to whether models can communicate complex medical information in ways that are understandable to patients without medical training. This direction is both socially impactful and highly relevant to real-world healthcare applications, where patient comprehension and informed decision-making depend on clear communication.
2. Robust Evaluation: The paper's evaluation is further bolstered by evaluating LLMs on three different pillars - linguistic simplicity, jargon density, and factual informativeness.  This ensures that the models are not only producing “easier” texts but are also preserving accuracy, which is critical in healthcare, where over-simplification can be dangerous. The inclusion of both open-source and proprietary models provides a balanced landscape view of the field.
3. The dataset is built and reviewed by professionals in nursing and public health, ensuring questions and answers reflect genuine patient information needs.

**Weaknesses:**

1. The paper repeatedly claims that REPQA is designed for lay users or patients without medical backgrounds, yet the chosen readability targets (Flesch–Kincaid grade levels 6, 9, 12, and 15) are never grounded in actual measures of patient literacy or health-communication standards. There is no justification for why a target score of 6  represents the intended audience. Moreover, the analysis misses an opportunity to relate model behavior at different grade levels to real-world comprehension gaps — for instance, how responses at FK=6 vs. FK=9 affect clarity for lay readers. Without this linkage, the results remain abstract and disconnected from the paper’s stated goal of serving lay audiences.
2. The paper reports extensive quantitative results (Tables 4–5) but fails to highlight key trends or outliers. Important comparative insights—such as which models best balance readability and accuracy—are buried in numbers without synthesis. A clearer narrative summarizing overall patterns is missing.
3. In-depth diagnostics are conducted mainly on open-source models, leaving proprietary systems under explored. This limits the generality of insights and obscures whether observed behaviors (e.g., undershooting readability targets) generalize across architectures.
4. The reinforcement signal optimizes only for readability, not correctness. As a result, the observed readability–accuracy trade-off is somewhat trivial, since accuracy was never part of the objective. This weakens claims about the inherent nature of the trade-off.
5. Some reference answers include value judgments (e.g., “the Mediterranean diet is healthy”), raising concerns about bias and consistency in the dataset’s gold standards.
6. Both Flesch–Kincaid and the professional-term ratio capture lexical simplicity but ignore semantic clarity and completeness. These metrics cannot detect whether essential qualifiers, side effects, or caveats were omitted—an especially serious issue in medical communication.
7. All evaluations rely on automated metrics or LLM-based critics. Without human judgment—either from patients or clinicians—it is impossible to confirm whether lower-grade outputs are truly more comprehensible or trustworthy.
8. Simplification often shortens answers, but the paper does not examine whether shorter responses omit critical details. Understanding this relationship is key to assessing whether improved readability compromises informativeness.
9. Moreover, while the paper reports overall QA accuracy, it lacks a content-granular evaluation—for example, distinguishing whether simplified outputs omit minor supporting details or major medical qualifiers. Without such analysis, it is impossible to determine whether readability gains merely remove redundant wording or meaningfully distort information, limiting the benchmark’s utility for assessing safe patient communication.

**Questions:**

1. Could you provide detailed statistics on inter-annotator agreement during the expert review process? What specific metrics were used, and how were disagreements resolved?
2. How do automated readability metrics (Flesch-Kincaid) correlate with actual patient comprehension and satisfaction? Have you conducted human evaluation studies with lay users rating clarity and trustworthiness?
3. To what extent does the multiple-choice QA task serve as an effective proxy for evaluating LLM-generated patient education content? Specifically, does the MCQA task cover the major topics present in the generated context, or are minor and related details overlooked? Human evaluation of coverage will be crucial to assess how well the MCQA task reflects the completeness and factual quality of the generated answers.
4. How sensitive are your findings to the choice of readability metrics? Would results differ significantly with alternative measures like SMOG?
5. How do you ensure that readability improvements don't compromise patient safety through oversimplification? Do simplified responses maintain critical medical disclaimers and safety warnings?
6. How might readability needs vary across different demographic groups (age, education, health literacy) and medical contexts (emergency vs. routine care)?

---

> ### Author Response · Authors · 2025-11-22
>
> ```
> C1: There is no justification for why a target score of 6 represents the intended audience.
> ```
> As recommended by the U.S. Agency for Healthcare Research and Quality (AHRQ), patient-facing health information should be written at approximately a 5th-grade reading level, which corresponds to a Flesch-Kincaid grade of ~6. This is why we adopt 6 as the target score for readability.
>
> https://www.ahrq.gov/sites/default/files/wysiwyg/professionals/quality-patient-safety/quality-resources/tools/literacy-toolkit/healthlittoolkit2_tool11.pdf
>
> ```
> C2: The author does not highlight insights from Table 4.
> ```
> Thanks for the suggestion. We made the following analysis.
>
> Among these models, Claude-3.7-Sonnet outputs responses with the highest average grade level, while Qwen2.5-32B achieves the lowest grade level, and LLaMA3.1-8B is the closest to our target value (i.e., 6). As for the professional score, BioMistral-7B produces the most specialized language, likely due to its fine-tuning on medical corpora. In contrast, Gemma3-12b yields the lowest professional word usage, which is preferable for readability. As for accuracy, o4-mini attains the highest score at 83.68%.
>
> Across the four evaluated categories (suggestion, fact, definition, rationale), the definition category is the easiest, with an average accuracy of 84.06%, whereas the fact category is the most challenging, with an average accuracy of 72.80%. Notably, the suggestion category consistently exhibits both the lowest Flesch-Kincaid grade level and professional score, which aligns with intuition: suggestions typically do not require complex language compared to the explanation of medical facts or rationales.
>
> Among proprietary models, we observe that more advanced versions within a series consistently demonstrate improved accuracy. For instance, accuracy increases with model upgrades such as o3-mini→o4-mini and GPT-4-turbo→GPT-4 → GPT-4.1. These trends indicate that newer model iterations are more effective in generating accurate responses. We have also incorporated the analysis in Appendix D.
>
> ```
> C3: In-depth diagnostics are conducted mainly on open-source models, leaving proprietary systems underexplored, which limits the generality of insights and obscures whether observations generalize across architectures.
> ```
> We have now included the GPT-4.1 results in Figure 3 as additional representative thinking models. Similar to other open-source methods, GPT-4.1 still struggles to produce answers at the target readability level and fails to differentiate between answers or questions with varying readability scores, which is consistent with our analysis.
>
> ```
> C4: The reinforcement signal optimizes only for readability, not correctness, which weakens the claim of the contradiction between readability and accuracy.
> ```
> In practice, the ground truth is usually absent in this domain. Therefore, our setting faithfully reflects what might occur in real-world deployments, where systems must optimize for readability without direct correctness supervision. Further, if the two objectives do not contradict each other, then optimizing readability—even in the absence of correctness signals—should not have harmed accuracy. The fact that substantial degradation *does* occur demonstrates that the trade-off is not an artifact of our training setup, but rather a genuine tension between making answers more accessible and preserving their factual integrity.
>
> ```
> C5: Some reference answers include value judgments (e.g., “the Mediterranean diet is healthy”).
> ```
> We appreciate the reviewer’s concern. We would like to clarify that statements such as “the Mediterranean diet is healthy” are not value judgments but reflect consensus findings from authoritative public health and nutritional guidelines (https://www.dietaryguidelines.gov/sites/default/files/2020-12/Dietary_Guidelines_for_Americans_2020-2025.pdf). These reference answers were drawn from reputable, evidence-based sources and reviewed by domain experts to ensure they convey established scientific understanding rather than subjective evaluations.

---

> > ### Author Response · Authors · 2025-11-22
> >
> > ```
> > C6: Both Flesch–Kincaid and the professional-term ratio capture lexical simplicity but ignore semantic clarity and completeness.
> > ```
> > We have included these two metrics (Dale Chall and Smog), as well as the Yngve depth, which is used to measure syntax complexity.
> >
> > | Model | kincaid-Flesch | Dale Chall | Smog | Yngve |
> > | --- | --- | --- | --- | --- |
> > | o4-mini | 3.79 | 7.03 | 5.53 | 2.84 |
> > | o3-mini | 3.99 | 6.72 | 5.56 | 2.55 |
> > | GPT-4o | 5.00 | 7.56 | 6.97 | 2.63 |
> > | GPT-4o-mini | 4.42 | 7.07 | 6.62 | 2.56 |
> > | GPT-4.1 | 4.31 | 6.74 | 5.93 | 2.68 |
> > | GPT-4 | 4.46 | 6.89 | 6.28 | 2.57 |
> > | GP-4-turbo | 5.18 | 6.69 | 5.97 | 2.66 |
> > | Claude-3.7-sonnet | 8.63 | 8.43 | 7.60 | 3.60 |
> > | Claude-3.5-sonnet | 6.50 | 7.59 | 6.44 | 2.99 |
> > | Gemini-1.5-pro | 4.73 | 6.87 | 6.82 | 2.63 |
> > | Gemini-2.0-flash | 4.12 | 6.45 | 6.06 | 2.53 |
> > | Gemini-1.5-flash | 4.25 | 7.10 | 6.47 | 2.44 |
> > | LLaMA-3.1-8B | 6.27 | 8.22 | 8.19 | 2.89 |
> > | Qwen2.5-7B | 4.49 | 6.59 | 5.72 | 2.57 |
> > | Phi-4 | 5.30 | 7.21 | 7.49 | 2.76 |
> > | Yi-1.5-9B | 8.12 | 8.56 | 8.96 | 2.98 |
> > | Mistral-7B | 6.91 | 9.22 | 7.55 | 2.84 |
> > | InternLM-8B | 6.72 | 8.31 | 7.60 | 3.05 |
> > | Gemma-3-12b | 3.54 | 7.87 | 6.69 | 2.49 |
> > | DeepSeek-V2-Lite | 6.89 | 8.29 | 8.30 | 2.82 |
> > | LLaMA-3.1-70B | 8.17 | 8.78 | 8.63 | 3.21 |
> > | Qwen2.5-14B | 5.76 | 7.35 | 6.55 | 2.73 |
> > | Qwen2.5-32B | 3.50 | 6.34 | 5.03 | 2.41 |
> > | Qwen2.5-72B | 3.68 | 6.73 | 5.05 | 2.49 |
> >
> > These metrics are all strongly correlated with Flesch–Kincaid (Dale–Chall: 0.800; SMOG: 0.806; Yngve Depth: 0.823), the primary readability measure used in the main text. The close agreement across metrics further supports the robustness of our conclusions.
> >
> > ```
> > C7: All evaluations rely on automated metrics or LLM-based critics. Without human judgment—either from patients or clinicians—it is impossible to confirm whether lower-grade outputs are truly more comprehensible or trustworthy.
> > ```
> > Our dataset construction and experimental design were supported by an interdisciplinary team with nursing and public health expertise (as noted in Table 8), ensuring the methodological soundness of our study. We acknowledge, that beyond in-silico evaluation, human evaluation is essential. Due to the limited rebuttal period, we cannot complete such evaluations at this stage, but we intend to incorporate a thorough human-subject assessment in our subsequent manuscript.
> >
> > ```
> > C8: Simplification often shortens answers, but the paper does not examine whether shorter responses omit critical details. Understanding this relationship is key to assessing whether improved readability compromises informativeness.
> > ```
> > We have already discussed this issue through two case studies (see Figure 4), where we compare the responses generated by the base model and the post-trained model using TA-GRPO. The results show that the model with higher readability (Qwen+TA-GRPO) indeed produces more concise answers; however, this simplification sometimes omits key information (Line 465), resulting in reduced accuracy, which highlights the significance of our proposed benchmark.
> >
> > ```
> > C9: While the paper reports overall QA accuracy, it lacks a content-granular evaluation—for example, distinguishing whether simplified outputs omit minor supporting details or major medical qualifiers. Without such analysis, it is impossible to determine whether readability gains merely remove redundant wording or meaningfully distort information.
> > ```
> > As discussed in C8, the two case studies presented in Figure 4 have demonstrated such a readability-accuracy trade-off, which is caused by the loss of the critical information. Specifically, the way Qwen+TA-GRPO improves readability is by reducing the use of medical terms (lower prof. term) and simplifying the explaination, such as list compression. Such simplification indeed lose critical information, and we believe one of the contributions of our study is to highlight such a trade-off and inspire more future studies.

---

> > > ### Author Response · Authors · 2025-11-22
> > >
> > > ```
> > > Q1: What metrics were used, and how were disagreements resolved in the expert review process?
> > > ```
> > > The review procedure followed national, evidence-based guidelines (Appendix A.2), but—as the task required qualitative domain judgment rather than categorical labeling—we did not employ a multi-annotator labeling setup designed for computing formal inter-annotator agreement metrics such as Cohen’s κ or Fleiss’ κ. Instead, the reviewers discussed each question individually, evaluating it *case by case* based on its specific context and content, and reached decisions through group consensus.
> > >
> > > ```
> > > Q2: Has authors conducted human evaluation studies with lay users rating clarity and trustworthiness?
> > > ```
> > > We agree that human evaluation is indispensable beyond our in-silico analyses for evaluating the readability. Given the constraints of the rebuttal period, we are unable to conduct such evaluations at this time, but we plan to include a comprehensive human-subject assessment in our subsequent manuscript.
> > >
> > > ```
> > > Q3: To what extent does the multiple-choice QA task serve as an effective proxy for evaluating LLM-generated patient education content? Specifically, does the MCQA task cover the major topics, or are minor and related details overlooked? Human evaluation of coverage will also be crucial.
> > > ```
> > > At this stage, our proxy MCQA is intentionally designed to capture the major facts in each reference answer rather than the more granular details. Focusing on core concepts allows us to measure factual accuracy in the simplest and most reliable way using a single multiple-choice item per question.
> > >
> > > We acknowledge, however, that minor details can also be clinically important. In future iterations, we plan to expand the proxy MCQA to include questions targeting secondary details and incorporate human evaluation.
> > >
> > > ```
> > > Q4: How sensitive are findings to the choice of readability metrics? Would results differ significantly with alternative measures like SMOG?
> > > ```
> > > As shown in **C6**, other metrics are all strongly correlated with Flesch–Kincaid (Dale–Chall: 0.800; SMOG: 0.806; Yngve Depth: 0.823), the primary readability measure used in the main text. The close agreement across metrics further supports the robustness of our conclusions.
> > >
> > > ```
> > > Q5: How do you ensure that readability improvements don't compromise patient safety through oversimplification? Do simplified responses maintain critical medical disclaimers and safety warnings?
> > > ```
> > > We would like to politely point out that our goal in this work is not to demonstrate state-of-the-art RL performance or to propose an optimal solution. Instead, we intentionally use a relatively simple solution to show that the readability–accuracy trade-off remains difficult to optimize. This difficulty is precisely what motivates our benchmark: by showing that current methods struggle with a straightforward RL algorithm, we provide evidence that the problem is inherently challenging and that our benchmark captures a meaningful and underexplored dimension of current LLMs.
> > >
> > > As discussed in C8 and C9, we also include a case study analyzing potential causes of reduced accuracy under improved readability. In future work, we plan to incorporate a comprehensive human evaluation involving clinical experts to assess whether safety-critical information is preserved and to ensure that readability improvements do not introduce trustworthiness or safety issues.
> > >
> > > ```
> > > Q6: How might readability needs vary across different demographic groups (age, education, health literacy) and medical contexts (emergency vs. routine care)
> > > ```
> > > We agree that readability needs can vary across demographic groups (e.g., age, education level, and health literacy) and across medical contexts. However, according to the U.S. Agency for Healthcare Research and Quality (AHRQ), to generally ensure accessibility for all patients (https://www.dietaryguidelines.gov/sites/default/files/2020-12/Dietary_Guidelines_for_Americans_2020-2025.pdf), we should make patient-facing materials generally be written at approximately a 5th-grade level.
> > >
> > > Therefore, our benchmark adopts this widely used standard to provide a unified evaluation target. Tailoring readability to specific subpopulations or clinical contexts is important but beyond the scope of the current study; we view it as a promising direction for future extensions of REPQA.

---

### Official Review · Reviewer_5RWL · 2025-11-10

**Soundness:** 3
**Presentation:** 3
**Contribution:** 3
**Rating:** 4
**Confidence:** 4

**Summary:**

This paper introduces RepQA, a benchmark dataset specifically designed to evaluate the readability of outputs generated by large language models (LLMs) in patient education question-answering scenarios.
Unlike previous work that focused solely on medical question-answering accuracy, REPQA emphasizes readability, i.e., whether LLMs can explain health issues to the general public using plain language.

**Strengths:**

1. Focusing on readability for patients, which addresses an important practical application dimension in medical LLM research.
2. Employing expert-reviewed QA pairs covering broad health topics, explicitly curated for “non-expert readers”;
3. Comprehensive evaluation, including many representative LLMs with many results.
4. Discussion on the trade-off between accuracy and readability, with RL.

**Weaknesses:**

1. It's confusing in Table 1 that the authors claim this benchmark as `Long Answer & Multiple Choice`, while in Table 2, the reader can only understand this as a multiple-choice QA benchmark.
2. Relying solely on the Flesch-Kincaid readability and the proportion of medical terms fails to capture deeper characteristics such as semantic conciseness, structural coherence, or syntactic complexity. Employing a combination of LLMs and human evaluation may be a more suitable approach.
3. Regarding RL, authors may not have found an optimal solution to address the trade-off between readability and accuracy. One reason for this is probably the previous point: is a reward model based on such a simple rule not so reliable?
4. Would the questions be too simple? Since the numbers in Table 4 are quite good.

**Questions:**

1. Consider using `large language model (LLM)`, not `Large Language Model`, like https://en.wikipedia.org/wiki/Large_language_model
2. Consider mark important numbers in Table 4.

---

> ### Author Response · Authors · 2025-11-22
>
> ```
> C1: Table 1 shows that the benchmark is long answer & multiple choice QA, while the reviewer can only understand it as multiple choice QA.
> ```
> The main focus of our benchmark is **long answer**. As illustrated in Figure 2’s blue box, the model’s task is to output a complete response. The multiple-choice QA, which is called proxy in our work (line 219), is used to evaluate the accuracy of the long answer, rather than testing the model’s performance on the patient education questions. We will emphasize it in the revised version.
>
> ```
> C2: The author should include metrics besides Flesch-Kincaid score and include human evaluations.
> ```
> We thank the reviewer for the suggestion. We propose to use Yngve depth score to capture syntactic complexity, as well as other extra metrics to capture readability.
>
> | Model | Grade | Dale Chall | Smog | Yngve |
> | --- | --- | --- | --- | --- |
> | o4-mini | 3.79 | 7.03 | 5.53 | 2.84 |
> | o3-mini | 3.99 | 6.72 | 5.56 | 2.55 |
> | GPT-4o | 5.00 | 7.56 | 6.97 | 2.63 |
> | GPT-4o-mini | 4.42 | 7.07 | 6.62 | 2.56 |
> | GPT-4.1 | 4.31 | 6.74 | 5.93 | 2.68 |
> | GPT-4 | 4.46 | 6.89 | 6.28 | 2.57 |
> | GP-4-turbo | 5.18 | 6.69 | 5.97 | 2.66 |
> | Claude-3.7-sonnet | 8.63 | 8.43 | 7.60 | 3.60 |
> | Claude-3.5-sonnet | 6.50 | 7.59 | 6.44 | 2.99 |
> | Gemini-1.5-pro | 4.73 | 6.87 | 6.82 | 2.63 |
> | Gemini-2.0-flash | 4.12 | 6.45 | 6.06 | 2.53 |
> | Gemini-1.5-flash | 4.25 | 7.10 | 6.47 | 2.44 |
> | LLaMA-3.1-8B | 6.27 | 8.22 | 8.19 | 2.89 |
> | Qwen2.5-7B | 4.49 | 6.59 | 5.72 | 2.57 |
> | Phi-4 | 5.30 | 7.21 | 7.49 | 2.76 |
> | Yi-1.5-9B | 8.12 | 8.56 | 8.96 | 2.98 |
> | Mistral-7B | 6.91 | 9.22 | 7.55 | 2.84 |
> | InternLM-8B | 6.72 | 8.31 | 7.60 | 3.05 |
> | Gemma-3-12b | 3.54 | 7.87 | 6.69 | 2.49 |
> | DeepSeek-V2-Lite | 6.89 | 8.29 | 8.30 | 2.82 |
> | LLaMA-3.1-70B | 8.17 | 8.78 | 8.63 | 3.21 |
> | Qwen2.5-14B | 5.76 | 7.35 | 6.55 | 2.73 |
> | Qwen2.5-32B | 3.50 | 6.34 | 5.03 | 2.41 |
> | Qwen2.5-72B | 3.68 | 6.73 | 5.05 | 2.49 |
>
> These metrics are all strongly correlated with Flesch–Kincaid (Dale–Chall: 0.800; SMOG: 0.806; Yngve Depth: 0.823), the primary readability measure used in the main text. The close agreement acros metrics further supports the robustness of our conclusions. The table is also updated in Table 10 in the manuscript.
>
> For human evaluation,  while our dataset construction and experimental design were supported by an interdisciplinary team with nursing and public health expertise (as noted in Table 8),  we acknowledge, that beyond in-silico evaluation, human evaluation is essential. Due to the limited rebuttal period, we cannot complete such evaluations at this stage, but we intend to incorporate a thorough human-subject assessment in our subsequent manuscript.
>
> ```
> C3: The reviewer is worried that the RL trade-off may be misleading because the reward model is overly simplistic and therefore may not reliably capture readability or accuracy.
> ```
> We would like to politely point out that our goal in this work is not to demonstrate state-of-the-art RL performance or to propose an optimal solution. Instead, we intentionally use a relatively simple solution to show that the readability–accuracy trade-off remains difficult to optimize. This difficulty is precisely what motivates our benchmark: by showing that current methods struggle with a straightforward RL algorithm, we provide evidence that the problem is inherently challenging and that our benchmark captures a meaningful and underexplored dimension of current LLMs.
>
> ```
> C4: Questions might be too simple since the accuracy in Table 4 is high.
> ```
> Instead of proposing challenging medical tasks, our work focuses on how well models can *communicate* that knowledge to lay users. Even so, accuracies are far from saturated (often in the 70–80% range, and sometimes near 65%), and Table 5 shows that once models are pushed to improve readability, their accuracy drops further, especially for Suggestion and Fact questions, indicating that the task still exposes substantial failure modes.
>
> ```
> Q1: Consider using `large language model (LLM)`, not `Large Language Model`
> ```
> Thanks for the suggestion. We have updated them in the revised version.
>
> ```
> Q2: Consider mark important numbers in Table 4.
> ```
> Thanks for the suggestion. We have highlighted the best performance of each column.

---

### Note · Program_Chairs · 2026-01-22
**Submission Desk Rejected by Program Chairs**

Pdf reveals author names.